# Drawing out of Distribution with Neuro-Symbolic Generative Models

**Yichao Liang**[1,2]**, Joshua B. Tenenbaum**[2]**, Tuan Anh Le**[*,3] **& N. Siddharth**[*,4]

[1]University of Oxford, [2]MIT, [3]Google, [4]University of Edinburgh

## Abstract

Learning general-purpose representations from perceptual inputs is a hallmark of human intelligence. For example, people can write out numbers or characters, or even draw doodles, by characterizing these tasks as different instantiations of the same generic underlying process—compositional arrangements of different forms of pen strokes. Crucially, learning to do one task, say writing, implies reasonable competence at another, say drawing, on account of this shared process. We present **D**rawing **o**ut **o**f **D**istribution (DooD), a neuro-symbolic generative model of stroke-based drawing that can learn such general-purpose representations. In contrast to prior work, DooD operates directly on images, requires no supervision or expensive test-time inference, and performs unsupervised amortised inference with a symbolic stroke model that better enables both interpretability and generalization. We evaluate DooD on its ability to generalise across both data and tasks. We first perform zero-shot transfer from one dataset (e.g. MNIST) to another (e.g. Quickdraw), across five different datasets, and show that DooD clearly outperforms different baselines. An analysis of the learnt representations further highlights the benefits of adopting a symbolic stroke model. We then adopt a subset of the Omniglot challenge tasks, and evaluate its ability to generate new exemplars (both unconditionally and conditionally), and perform one-shot classification, showing that DooD matches the state of the art. Taken together, we demonstrate that DooD does indeed capture general-purpose representations across both data and task, and takes a further step towards building general and robust concept-learning systems.

## 1 Introduction

Humans can learn representations of data that are general-purpose and meaningful. Being general-purpose permits effective reuse when characterizing novel observations, and being meaningful facilitates tasks like generating or classifying observations. Key to this is a generic process for characterizing observations—inferring *what* features are relevant and *how* they compose to generate the observations [9, 13, 33]. For example, when observing handwritten numbers, we learn to characterise them as sequential compositions (*how*) of different pen strokes (*what*). This is general-purpose as it allows characterizing novel observations, say doodles instead of numbers, simply as novel compositions of previously learnt pen strokes. It is also meaningful since pen-strokes themselves are symbolic and interpretable.

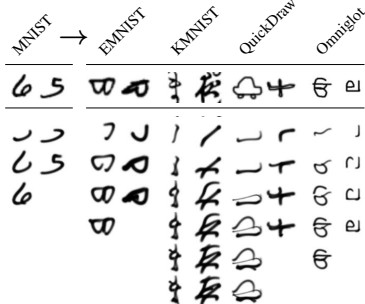

Figure 1: DooD trained on MNIST generalises to other data with no extra training. Each column denotes a target and its step-by-step reconstruction.

Current computational approaches capture important aspects of generalizability, but none of these is simultaneously efficient, reliable, interpretable, and unsupervised [25]. At one end, symbolic approaches like Lake et al. [24] attribute generalization to an explicit hierarchical composition process involving sub-strokes, strokes, and characters, and build concomitant models that demonstrate human-like generalization abilities across different tasks. At the other end, neural approaches like deep generative models [9, 18, 29] and deep meta-learning [11, 32, 34] favour scalable learning from raw

---

*equal contribution

36th Conference on Neural Information Processing Systems (NeurIPS 2022).

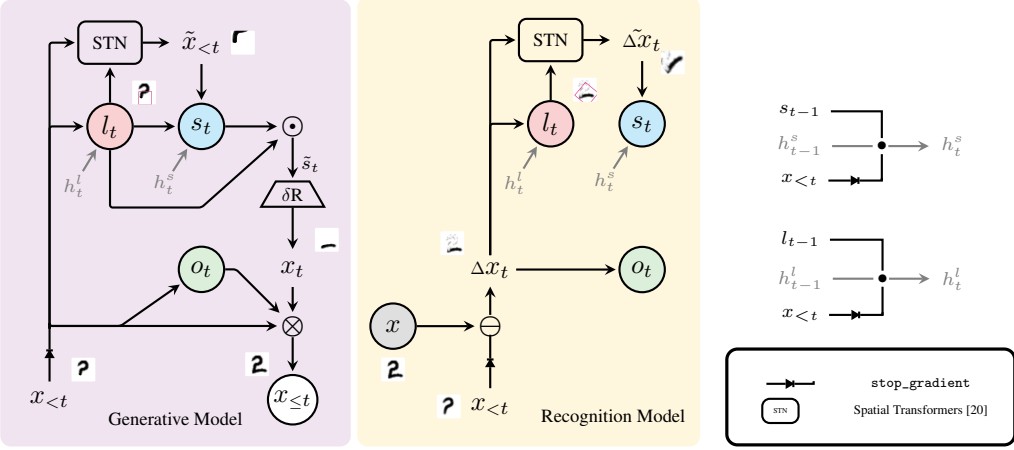

Figure 2: The *generative model* sequentially samples both an image location ($l_t$) and a corresponding stroke ($s_t$) at that location. The rendered stroke $x_t$ is composited onto intermediate rendering $x_{<t}$ to produce $x_{\leq t}$. A binary on/off variable ($o_t$) determines when to continue drawing. Differentiable rendering ($\delta \bar{R}$) and differentiable affine transformations via Spatial Transformer Networks (STNs) [20] enables gradient-based learning. The *recognition model* conditions on a residual $\Delta x_t = x - x_{<t}$ to sample where to draw next ($l_t$) and what to draw next ($s_t$), and whether to continue drawing ($o_t$). Both models are autoregressive via two (shared) RNNs with hidden states $h_t^s$ and $h_t^l$.

perceptual data, unfettered by explicit representational biases such as strokes and their compositions. Each comes with its own shortcomings—symbolic approaches typically need additional supervision or data processing along with expensive special-purpose inference, and neural approaches fail to generalise well and don't capture interpretable representations. Neuro-symbolic approaches [10] seek to make the best of both worlds by judiciously combining neural processing of raw perceptual inputs with symbolic processing of extracted features, but typically involve a different set of trade-offs.

We present **D**rawing **o**ut **o**f **D**istribution (DooD), a neuro-symbolic generative model of stroke-based drawing that can learn general-purpose representations (Fig. 1). Our model operates directly on images, requires no supervision, pre-processing, or expensive test-time inference, and performs efficient amortised inference with a symbolic stroke model that helps with both interpretability and generalization, setting us apart from the current state-of-the-art in neuro-symbolic approaches [10, 17]. We evaluate on two axes (a) generalization across data, which measures how well the learnt representations can be reused to characterise out-of-distribution data, and (b) generalization across task, where we measure how useful the learnt representations are for auxiliary tasks drawn from the Omniglot challenge set [24]. We show that DooD significantly outperforms baselines on generalization across datasets, highlighting the quality of the learnt representations as a factor, and on generalization across tasks, show that it outperforms neural models, while being competitive against SOTA neuro-symbolic models without requiring additional support such as supervision or data augmentation.

## 2 Method

The framework for DooD involves a generative model over sequences of strokes and their layouts, a recognition model that conditions on a given observation to predict where to place what strokes, and an amortised variational-inference learning setup that uses these models to estimate an evidence lower bound (ELBO) as the objective.

### 2.1 Generative Model

Conceptually, the model can be seen as drawing a figure over a sequence of steps, building up to the final image (as seen in Fig. 1). At each step $t$ the model identifies a region of the image canvas to draw in, puts down Bézier curve control points within that region, renders the curve in a differentiable manner, and then composites this rendered stroke over the previously rendered canvas. We refer to these as the *layout*, *stroke*, *rendering*, and *compositing* modules respectively (elaborated below).

The model sits on a substrate of recurrent neural networks (RNNs), one each for the layout and stroke modules, with hidden states $h_t^l$ and $h_t^s$ respectively. The complete setup is depicted in Fig. 2 along with example values (images) for the different variables involved. See Appendix B for details.

Formally, the generative model defines a joint distribution over an image $x$ composed over $T$ steps, with latent variables $l_t$ and $s_t$ that characterise a stroke's location and form, and a binary latent variable $o_t$ that determines stopping, as

$$p(x, l_{\leq T}, s_{\leq T}, o_{\leq T+1})$$

$$= \left( \prod_{t=1}^{T} p_{\text{on}}(o_t|x_{<t})\, p_{\text{stroke}}(s_t|l_t, \tilde{x}_{<t}, h_t^s)\, p_{\text{layout}}(l_t|x_{<t}, h_t^l) \right) p_{\text{comp}}(x|x_{\leq T})\, p_{\text{on}}(o_{T+1}|x_{\leq T}), \quad (1)$$

where terminal step $T$ is the last $t$ where $o_t = 1$, $x_t$ the rendered stroke given $(s_t, l_t)$, $x_{<t}$ the canvas-so-far, and all variables at step $t = 0$ initialised to $\mathbf{0}$. Note that the likelihood is evaluated only at the terminal step $T$.

**Layout Module:** At step $t$, given the canvas-so-far $x_{<t}$ and corresponding layout-RNN hidden state $h_t^l$, we define the layout as a distribution over affine transforms. This allows transforming the canvas into a "glimpse" $\tilde{x}_{<t}$ using a Spatial Transformer Network (STN) [20], which allows focussing on a particular canvas region. The affine transform is constructed from appropriately constrained scale ($l_t^{\text{sc}}$), translation ($l_t^{\text{tr}}$), and rotation ($l_t^r$) random variables, by employing a Gaussian Mixture Model (GMM) with $M$ components over the collection as

$$p_{\text{layout}}(l_t|x_{<t}, h_t^l) = \sum_{m=1}^{M} \alpha_m \cdot \mathcal{N}_{\text{sc},m}(l_t^{\text{sc},m}|x_{<t}, h_t^l) \cdot \mathcal{N}_{\text{tr},m}(l_t^{\text{tr},m}|x_{<t}, h_t^l) \cdot \mathcal{N}_{r,m}(l_t^{r,m}|x_{<t}, h_t^l), \quad (2)$$

$$\tilde{x}_{<t} = \text{STN}(l_t, x_{<t}).$$

**Stroke Module:** Given the sampled affine transform $l_t$, selected "glimpse" $\tilde{x}_{<t}$, and corresponding stroke-RNN hidden state $h_t^s$, this module defines a distribution over strokes parametrised as $D^{\text{th}}$ order Bézier splines, constructing a GMM with K components for each spline control point as

$$p_{\text{stroke}}(s_t|l_t, \tilde{x}_{<t}, h_t^s) = \prod_{d=0}^{D} \sum_{k=1}^{K} \pi_{d,k} \cdot \mathcal{N}_{\text{d,k}}(s_t^{d,k}|l_t, \tilde{x}_{<t}, h_t^s). \quad (3)$$

**Rendering Module:** Note that the Bézier spline control points sampled from the stroke module are taken to be in a canonical centered coordinate frame. While this can help simplify learning by reducing the variation required to be captured by the stroke module, the points can't be rendered onto the canvas as is. We situate them properly within the context of the previously determined "glimpse" by simply applying the affine transform $l_t$ to the control points as $\tilde{s}_t = l_t \odot s_t$. The transformed control points now describe a stroke to be drawn over the whole canvas for step $t$, through a differentiable renderer $\delta R$, to produce the rendered stroke as $x_t = \delta R(\tilde{s}_t)$ (see Appendix B.1 for details). Using the intermediate canvas to guide model unrolling is termed *execution guidance* (EG).

**Compositing Module:** At step $t$ this defines a distribution over whether the model should continue drawing strokes given the rendered canvas-so-far $x_{<t}$ as

$$p_{\text{on}}(o_t|x_{<t}) = \text{Bernoulli}(o_t|x_{<t}). \quad (4)$$

Once the model has decided to stop drawing, it stops permanently. At step $t$, the rendered stroke $x_t$ is composited with the canvas-so-far $x_{<t}$ to generate $x_{\leq t} = x_{<t} \otimes x_t$. When the model stops, i.e., at the last $t$ when $o_t = 1$ (denoted $T$), the likelihood, with global learnable parameter $\sigma$, is given as

$$p_{\text{comp}}(x|x_{\leq T}) = \text{Laplace}(x|x_{\leq T}, \text{diag}(\sigma)). \quad (5)$$

## 2.2 Recognition Model

As with prior approaches, we construct an approximate posterior to facilitate learning with amortised variational inference. Using the same notation from the generative model section, we define

$$q(l_{\leq T}, s_{\leq T}, o_{\leq T+1}|x) = q_{\text{on}}(o_{T+1}|\Delta x_{T+1}) \prod_{t=1}^{T} q_{\text{layout}}(l_t|\Delta x_t, h_t^l) q_{\text{stroke}}(s_t|\tilde{\Delta x}_t, h_t^s) q_{\text{on}}(o_t|\Delta x_t). \quad (6)$$

A couple of things stand out. First, where the generative model made heavy use of the canvas-so-far $x_{<t}$, the recognition model primarily uses the *residual* $\Delta x_t = x - x_{<t}$. Second, being given

the target observation $x$ itself, the information available to the layout and stroke modules is quite different. While they needed to speculate where, and what stroke to draw, in the generative model, here their task is simply to isolate a part of the drawing ("glimpse") and fit a spline to that.

As a consequence of these different characteristics, the distributions over layout and strokes in the recognition model do not need to be as flexible as the generative model—locating a curve in the residual and fitting it with a spline does not typically involve much ambiguity. To factor this in, and have the variational objective be reasonable, we define corresponding distributions in the recognition model using just a single component of the corresponding GMMs in the generative model as

$$q_{\text{layout}}(l_t|\Delta x_t, h_t^l) = \mathcal{N}_{\text{sc}}(l_t^{\text{sc}}|\Delta x_t, h_t^l) \cdot \mathcal{N}_{\text{tr}}(l_t^{\text{tr}}|\Delta x_t, h_t^l) \cdot \mathcal{N}_r(l_t^r|\Delta x_{<t}, h_t^l), \tag{7}$$

$$\tilde{\Delta} x_t = \text{STN}(l_t, \Delta x_t),$$

$$q_{\text{stroke}}(s_t|\tilde{\Delta} x_t, h_t^s) = \prod_{d=0}^{D} \mathcal{N}_{\text{d}}(s_t^d|\tilde{\Delta} x_t, h_t^s). \tag{8}$$

### 2.3 Learning

Having defined the generative and recognition models, we now bring them together in order to construct the variational objective that will enable learning both models simultaneously from data.

$$\log p(x) \geq \mathbb{E}_{q(l_{\leq T}, s_{\leq T}, o_{\leq T+1}|x)} \left[ \log \frac{p(x, l_{\leq T}, s_{\leq T}, o_{\leq T+1})}{q(l_{\leq T}, s_{\leq T}, o_{\leq T+1}|x)} \right] \tag{9}$$

Note that except for the stopping criterion $o_t$ which is a Bernoulli random variable, all other distributions employed are reparametrizable. In order to construct an effective variational objective with this discrete variable, we employ a control variate method, NVIL [28], that helps reduce the variance of the standard REINFORCE estimator, as is also done in related work [9] (see Appendix C.2).

Furthermore, in order to ensure that the ELBO objective is appropriately balanced, we employ additional weighting $\beta$ for the KL-divergence over stopping criterion $o_t$ within the objective [2, 19] (see Appendix C.1 for details). This weight plays a crucial role as a mismatch could result in the model either stopping too early or too late, resulting in incomplete or incorrect figures respectively.

## 3 Experiments

We wish to understand how well DooD generalises across both datasets (§ 3.1) and tasks (§ 3.2). For across-dataset generalization, we train DooD and Attend-Infer-Repeat (AIR) [9][2], an unsupervised part-based model, on each of five stroke-based image datasets (i) MNIST (handwritten digits) [26], (ii) EMNIST (handwritten digits and letters) [7], (iii) KMNIST (cursive Japanese characters) [6], (iv) Quickdraw (doodles) [15], and (v) Omniglot (handwritten characters from multiple alphabets) [24], and evaluate how well the model generalises to unseen exemplars both within the same dataset and across other datasets. We find that DooD significantly outperforms AIR, which from ablation studies, is attributed to explicit stroke modelling and execution guidance. Note that we only compare against a fully-unsupervised approach since most datasets do not provide additional data in the form of stroke labels (as required elsewhere [10]). For across-task generalization, we primarily focus on Omniglot and evaluate on three out of the five challenge tasks for this dataset [25], which include contextual generation and classification. We find that our model outperforms unsupervised baselines, and is competitive against SOTA neuro-symbolic models without requiring additional support in the form of supervision or data augmentation. We include exact details about datasets (Appendix A), our model and the baselines (Appendix B), the training procedure (Appendix C), and the evaluation procedure (Appendix D) in the supplementary material.

### 3.1 Across-Dataset Generalization

**MNIST-trained transfer.** To understand how our model and AIR generalise to new datasets, we look at sequential reconstructions (Fig. 3). We train on MNIST and show sample reconstructions from all five datasets without fine tuning. Each model renders one step at a time by rendering latent parses of increasing length, allowing us to evaluate and compare the performance of part decomposition and inference. Note that we limit the maximum number of strokes to 6 throughout all experiments.

---

[2]specifically Difference-AIR, which uses execution guidance and performs much better than vanilla AIR

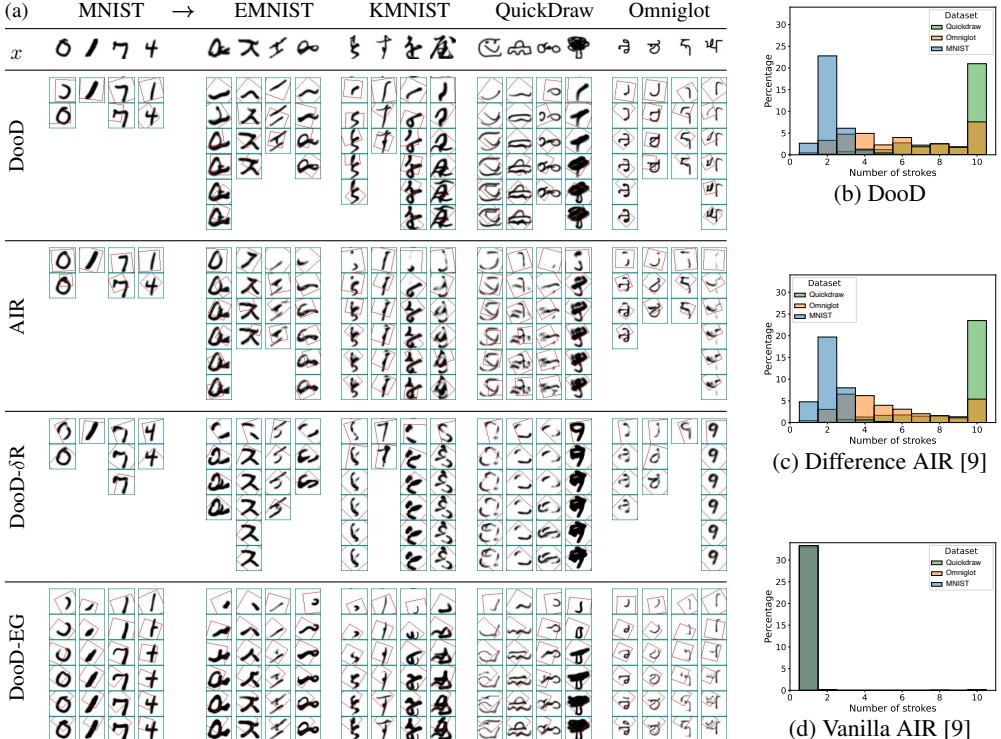

Figure 3: (a) Our model generalises better than AIR. Our model trained on MNIST reconstructs characters from all other four datasets while the baseline AIR model's reconstructions are often inaccurate, blurry or incomplete. Explicit stroke parametrization ($\delta$R) and execution-guided inference (EG) are responsible for this generalization which degrades when using our model without either of these components. (b-d) Both DooD and Difference AIR (AIR elsewhere) trained on MNIST generalise to using more strokes unlike Vanilla AIR which doesn't have execution-guided inference.

Our model reconstructs in-distribution images perfectly and out-of-distribution images near-perfectly while using fewer strokes for simpler datasets (e.g. MNIST) and more strokes for more complex datasets (e.g. Omniglot). While the AIR baseline also uses an appropriate number of steps for more complex datasets, the reconstructions degrade significantly for out-of-distribution images—they are blurry (e.g. the car & motorbike in QuickDraw), strokes go missing (e.g. the second KMNIST image) or the reconstructions are inaccurate (e.g. the last Omniglot character).

**Ablation studies.** To better understand why our model generalises well, we evaluate two further variants of DooD that ablate a key component each: an explicit spline decoder (DooD-$\delta$R) and execution-guided inference (DooD-EG) (Fig. 3a). In the model without an explicit spline decoder, we replace the differentiable spline renderer by a neural network decoder similar to AIR. This model still differs from the AIR in terms of the learnable sequential prior and the fact that we enforce explicit constrains over the latent variable ranges—e.g. enforcing the mean of the control-point Gaussian to not stray too far away from the image frame. In the variant without execution guidance, we do not perform intermediate rendering, removing the direct dependence of the generative model and the recognition model on the canvas-so-far $x_{<t}$ and the residual $\Delta x_t$.

Both the explicit spline decoder and the execution guidance prove to be important. Without the explicit spline decoder (DooD-$\delta$R), the reconstruction quality suffers—the strokes are blurry (e.g. first three QuickDraw images), strokes go missing (e.g. the last EMNIST image), or the reconstructions are inaccurate (e.g. the last Omniglot character is interpreted as a "9" due to overfitting). However, even without the explicit spline decoder, the model learns to be parsimonious, using fewer strokes to reconstruct simpler images (Fig. 3b-d). On the other hand, without execution guidance (DooD-EG), the model is unable to be selective with the number of strokes, always using the maximum allowed number. And while the reconstructions are better than AIR and DooD-$\delta$R, it still shows instances of missing strokes (e.g. some Omniglot characters). Note that although we use the canvas-so-far in a manner that disallows gradients (`stop_gradient`), just providing it as a conditioning variable for the different components (layout, stroke, RNN hidden states) has a tangible effect.

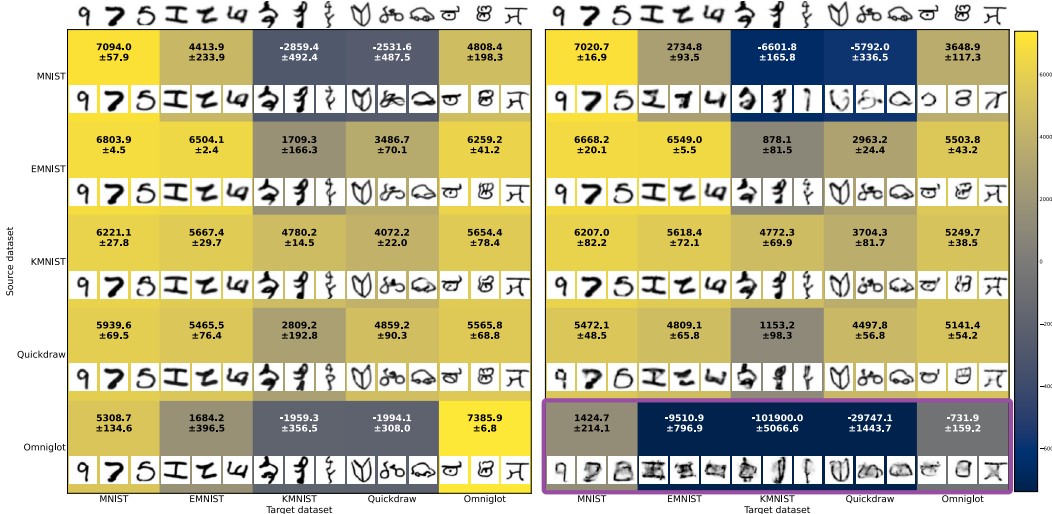

Figure 4: When training on a "source" dataset and testing on another "target" dataset, our model, DooD, (left) has a higher log marginal likelihood (values in each cell) than AIR (right). Given targets on top of the tables, DooD's reconstructions (images in each cell) are high quality when transferring out of distribution, unlike AIR which often struggles. Training on MNIST or Omniglot as a source dataset leads to worse transfer (the corresponding rows are the darkest) due to a larger distribution shift. Particularly AIR fails when trained on Omniglot using a Laplace likelihood (standard across all other model-dataset combination for good reconstructions), due to which we employ a Gaussian likelihood just for Omniglot-trained AIR (highlighted in purple).

**Quantifying zero-shot transfer.** We look at how DooD and AIR trained on each of the five datasets transfers to each other dataset to further understand how our model generalises. Models are trained on each "source" dataset and tested on each "target" dataset, resulting in a $5 \times 5$ table for each model (Fig. 4). Each cell shows the log marginal likelihood of the target dataset using the model trained on the source dataset, estimated using the importance weighted autoencoder (IWAE) objective [3] with 200 samples (mean and standard deviation over five runs). We also show reconstructions obtained by running the model trained on the source dataset on a few examples from the target dataset.

Our model generalises significantly better than AIR across datasets (off diagonal cells), while also performing better within dataset (diagonal cells). For both models, the values on the diagonal are the highest in any given column, suggesting that not training on directly on the target dataset results in a worse performance, as expected. For both models, the row values for MNIST and Omniglot are lower than in other rows, indicating that transfer learning performance is the worst when the source dataset is MNIST or Omniglot—potentially due to a larger distribution shift since MNIST has low diversity and Omniglot has little to no variation in stroke thickness, in contrast to the other datasets. However, we note that our reconstructions are high quality despite transferring out of distribution, unlike reconstructions from AIR which are qualitatively worse. For example, when transferring from simple datasets (MNIST), AIR makes incomplete, incorrect and blurry reconstructions, as we have seen before, while AIR trained on complex datasets like Omniglot results in blurry reconstructions for both in-distribution and out-of-distribution datasets. Furthermore, AIR fails when trained on Omniglot using a Laplace likelihood (used as standard across all other model-dataset combination). We thus employ a Gaussian likelihood just for Omniglot-trained AIR, and highlight it as an outlier.

**Understanding learned representations.** To better understand DooD's generalization ability, we investigate its learnt representations by clustering the inferred strokes using $k$-means clustering ($k = 8$), and study the clusters both qualitatively and quantitatively. For AIR, we cluster the corresponding part-representation latents. We then visualise things, using a t-SNE plot (Fig. 5) of the clusters, with exemplar strokes overlaid. We find that DooD has better-clustered representations, with clusters denoting largely distinct types of strokes—e.g., clusters for a "/", "c", and its horizontally flipped version. In contrast, the clusters from AIR are less sensible with some clusters even capturing full characters ("0"), comprising multiple strokes. There are also clusters which contain visually different strokes, and many visually similar strokes are assigned to different clusters. Quantitatively,

following Aksan et al. [1], we found DooD's better cluster consistency is reflected in a higher Silhouette Coefficient [30] than AIR (0.21 for DooD, 0.11 for AIR).

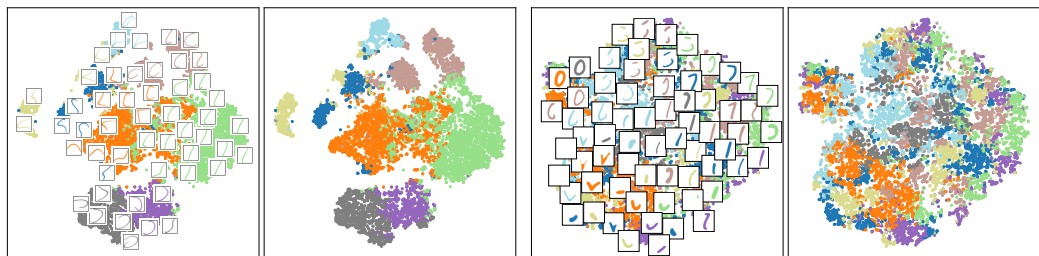

Figure 5: Clusters of inferred strokes for DooD (left) and inferred part representations for AIR (right) overlaid on a t-SNE plot. DooD's representation clusters to more semantically meaningful parts as indicated by better formed clusters.

## 3.2 Across-Task Generalization

Here, we focus on a subset of the Omniglot challenge tasks [24], to evaluate our model's utility on a range of auxiliary tasks that it was not trained to do. Despite much progress in deep generative modelling, relevant models are still not fully task general and often result in unrealistic (e.g. blurry) samples [10, 25]. DooD combines handling raw perceptual inputs with the compositional structure of strokes, which we evaluate on three out of the five Omniglot challenge tasks: unconditional generation, conditional generation, and one-shot classification. We compare against AIR and a state-of-the-art neuro-symbolic model (GNS [10]), where relevant. Note that GNS requires stroke and character-class supervision and practically, at least for now, only applies to Omniglot.

**Unconditional generation.** DooD generates realistic unconditional samples of all datasets (Fig. 6), indicating that the model has learned the high-level characteristics of each dataset. The strokes are sharp, and the stroke structure composes into realistic images from each dataset. For example, there are clear digits in the MNIST samples, there are recognizable objects (cars, bicycles, glasses, and smileys) in the QuickDraw samples, and the samples for EMNIST, KMNIST and Omniglot can be easily recognised as possible instances coming from those datasets. It generates samples of comparable fidelity to GNS without requiring any supervision, and as evaluated using the Fréchet inception distance (FID) [16] (smaller is better), outperforms GNS (0.051 versus 0.133).

The key to being able to generate realistic prior samples is the learnable sequential prior and the symbolic latent representation. AIR doesn't have a sequential prior, so although it is possible to get good reconstructions, it is impossible for it to generate realistic unconditional samples.

**Character-conditioned generation.** In order to generate new exemplars of the same Omniglot character, we follow Feinman and Lake [10], Lake et al. [24] and extend our model to a hierarchical generative model of an abstract character "type" or "template" that generates a concrete instance of a character "token", which is rendered out to an image. We consider the previously used latent variables as the type latent variable and introduce a token model which conditions on the type latent variable. The token model introduces (i) a drawing noise represented by adding a Gaussian noise with fixed standard deviations to spline control points and (ii) and an affine transformation on the noise perturbed points, whose parameters are also sampled from a Gaussian distribution (described in Appendix B.4). To sample a new exemplar of a character, we first sample the type variable from

| MNIST | EMNIST | KMNIST | QuickDraw | Omniglot (DooD) | Omniglot (GNS[10]) | Omniglot (True) |
|---|---|---|---|---|---|---|
| $.134 \pm .013$ | $.137 \pm .006$ | $.123 \pm .020$ | $.084 \pm .009$ | $.051 \pm .007$ | $.133 \pm .007$ | $.025 \pm .004$ |

Figure 6: DooD generates high quality unconditional character samples for all datasets which are visually indistinguishable from the real characters as it successfully captures the layout of strokes and their forms. Omniglot samples are compared to GNS [10] and real samples. Numbers denote Fréchet inception distance (FID), with smaller being better (mean $\pm$ 1 std. over 5 runs).

Figure 7: Given a target image of a handwritten Omniglot character, our model produces realistic new exemplars by inferring an explicit stroke-based representation.

Figure 8: Given a partially drawn character, our model can generate a realistic distribution over its completions by sampling from the generative model conditioned on the image of the partial character.

our recognition model, and produce different exemplars by sampling and rendering different token variables given this type variable. Distinctly from [10, 24], the parameters of the token model are not learned through supervision but simply set to a sensible value by examining the noise level in the output – any learned statistics can be straightforwardly plugged in.

DooD generates realistic new exemplars of complex QuickDraw drawings and Omniglot characters (Fig. 7) thanks to the accurate inference and the ability to add noise to explicitly parametrised strokes. While we can add an equivalent token model for AIR by (i) adding a Gaussian noise to the uninterpretable feature vector representing each part and (ii) applying a Gaussian affine transformation to the rendered image, the new exemplars are not as realistic both because of worse inference and the hard-to-control variations of the part vectors. GNS generates realistic conditional samples, but notably still makes unnatural samples in multiple instances (e.g., in column 1, 3 the detachments of strokes) despite having a hierarchical model learned with multi-levels of supervision.

**Partial completion.** As with inferring an entire figure in the previous case, we can interpret conditional generation in a slightly different way as well—where the condition is an *initialization* of a number, character, or figure, and the model tries to extend/complete it as best it can (Fig. 8). To do this, we first employ the recognition model over the partial figure to compute the hidden states of the shared recurrent networks. Next, starting with these computed states, we set the canvas-so-far $x_{<t}$ to be the partial figure itself and then unroll the generative model from that point onwards. As can be seen in the figure, DooD can generate a varied range of completions for each initial stroke, demonstrating its versatility and the utility of its learnt representations.

**One-shot classification.** Finally, we can apply the type-token hierarchical generative model used for generating new exemplars to perform within-alphabet, 20-way one-shot classification. The key quantity needed for performing this task is the posterior predictive score of the query image $x^{(T)}$ given a set of support images $\{x^{(c)}\}_c$, $\hat{c} = \arg\max_c p(x^{(T)}|x^{(c)})$, which requires marginalizing over the token variables corresponding to $x^{(T)}$ and $x^{(c)}$, and the type variable of $x^{(c)}$. Following [10, 24], we approximate this score by sampling from the recognition model given $x^{(c)}$, and perform gradient-based optimization to marginalize out the token variable of $x^{(T)}$ (details in Appendix D.2.3). We find that DooD outperforms the neural baseline (AIR), while attaining a competitive accuracy in comparison to other baselines (Table 1)

Table 1: Accuracy in one-shot classification, without data augmentation (DA), extra supervision (ES), or 2-way classification (2W).

| Model | DA | ES | 2W | Accuracy |
|---|---|---|---|---|
| AIR | ✗ | ✗ | ✗ | 14.5% |
| DooD | ✗ | ✗ | ✗ | 73.5% |
| VHE[18] | ✓ | ✓ | ✗ | 81.3% |
| GNS[10] | ✗ | ✓ | ✓ | 94.3% |
| BPL[24] | ✗ | ✓ | ✓ | 96.7% |

without requiring additional forms of support such as data augmentation, supervision for strokes, or more complex ways of computing the classification score such as two-way scoring (predicting $\hat{c} = \arg\max_c \frac{p(x^{(c)}|x^{(T)})}{p(x^{(c)})} p(x^{(T)}|x^{(c)})$).

## 4 Related Work

Our work takes inspiration from Lake et al. [24]'s symbolic generative modelling approach which hypothesises that the human ability to generalise comes from our causal and compositional understanding that characters are generated by composing substrokes into strokes, strokes into characters and rendering characters to images. As a result, Lake et al. [24] demonstrate human-like generalization on a wide range of tasks. However, it is trained using stroke sequences, and inference is performed using expensive MCMC sampling.

We combine features of neuro-symbolic generative models and deep generative models to be able to generalise well across tasks while using amortised inference and being unsupervised. From neuro-symbolic models, we share key features of Feinman and Lake [10]'s model like (i) using the canvas-so-far in the generative model and adopt a similar feature in the recognition model like Ellis et al. [8], (ii) parametrizing parts as splines and using a differentiable spline renderer, (iii) extending the model to have a type-token hierarchy for generating new exemplars and performing one-shot classification. Our model can be seen as an extension of [10] that learns directly from images and uses a recognition model for amortised inference. Like Hewitt et al. [17], we learn how to infer a stroke sequence directly from images using a differentiable renderer but infer strokes directly instead of learning a stroke bank and use a more flexible parametrization of strokes based on a differentiable spline renderer, leading to a more accurate model.

Similar to deep generative modelling approaches like [9, 14, 23, 29], we use attention to focus on parts of the canvas we want to generate to or recognise, and more generally exploit the compositionality of objects from which allows neural networks to learn simpler and hence more generalizable mappings. To be able to train our model from unsupervised images, we adopt the NVIL control variate [28] used by Eslami et al. [9] to be able to train a model with a discrete stop-drawing latent variable. This family of models, along with deep meta-learning approaches [11, 32, 34], is easier to learn due to the lack of symbolic variables and results in a fast amortised recognition model. However, the lack of strong inductive biases leads to poor and unreliable generalization [25]. We also share idea with other works combining deep learning and explicit stroke modelling [1, 12, 15, 27], but we focus on learning a principled generative model which allows tackling tasks like one-shot classification and generating new exemplars, in addition to conditional and unconditional sampling.

## 5 Conclusion

We demonstrated that DooD generalises across datasets and across tasks thanks to an explicit symbolic parametrization of strokes and execution guidance. This allows us to train on one dataset such as MNIST and generalise to a more complex, out-of-distribution dataset such as Omniglot. Given a compositional representation and an associated learned sequential prior, DooD can be applied to additional tasks in the Omniglot challenge like generating new exemplars and one-shot classification by extending it to have a type-token hierarchy. Our model produces realistic new exemplars without blur and artefacts unlike deep generative models.

More broadly, DooD is an example of a system that successfully combines symbolic generative models to achieve generalization and deep learning models to handle raw perceptual data and perform fast amortised inference while being learned from unsupervised data. We believe these principles can be useful for building fast, reliable and robust learning systems going beyond stroke-based data.

**Acknowledgements**  We thank Rob Zinkov for his contribution to the early versions of the renderer; Reuben Feinman, Luke Hewitt, Kevin Ellis for their thoughtful comments on the manuscript; and Reuben's prompt support in running the GNS model. NS was in part supported by the Turing 2.0 'Enabling Advanced Autonomy' project, funded by the EPSRC and the Alan Turing Institute.

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
