# Supplement to: Drawing out of Distribution with Neuro-Symbolic Generative Models

## A  Dataset Details

All dataset images are scaled to 50x50 in grayscale, with dataset-specific configuration list below.

**MNIST, KMNIST:** we use the original split of with 60k images for training, 10k for tests. Each image belongs to 1 of the 10 classes.

**EMNIST:** we use the "balanced" split of the dataset with 112,800 training images and 18,800 testing images, separated into 47 classes.

**QuickDraw:** we use the 10-category version of dataset, where each has 4k/1k training/testing samples, as found on `https://github.com/XJay18/QuickDraw-pytorch`.

**Omniglot:** we use the original split [24], with inverted black and white pixels. For one-shot classification (§ 3.2), we use the original task-split, as found on `https://github.com/brendenlake/omniglot`. It has 20 episodes, each a 20-way, 1-shot, within-alphabet classification task.

## B  Model Details

### B.1  Differentiable Renderer

Bézier curves are parametric curves commonly used in computer graphics to define smooth, continuous curves. The renderer outputs a greyscale, pixel-based image when takes in a stroke $\tilde{s}_t$ defined as an array of control-point coordinates for a Bézier curve. It has three parameters $(\sigma^t, \omega_0^t, \omega_1)$; the first two are per-stroke and $\omega_1$ is per-character. It renders a stroke through two steps: 1) compute a *sample curve* based on the control points; 2) rasterize the output image from the *sample curve* as described below.

With $(D+1)$ control points, $\tilde{s}_t = [p_d^t]_{d=0}^D$, each with their $x, y$ coordinates, a sample curve with $S$ sample points can be computed using the explicit definition of Bézier curves, where $n$ is one of the $S$ numbers ranging $[0, 1]$:

$$\mathbf{b}_n^t = \sum_{d=0}^D \binom{D}{d}(1-n)^{D-j}n^d p_d \tag{10}$$

In our case, we use $S = 100$ samples spaced evenly between $[0, 1]$.

With $S$ points $[\mathbf{b}_n^t]_{n=1}^S$, an image $\tilde{\tilde{x}}_t$, with pixels indexed by indexed by $h \in [0, H-1], w \in [0, W-1]$ where $H, W$ are image dimensions, can be rasterized. Its pixel intensity $\tilde{\tilde{x}}_{hw}^t$ is given by:

$$\tilde{\tilde{x}}_{hw}^t = \sum_n \frac{(h - \mathbf{b}_{n,x}^t)^2(w - \mathbf{b}_{n,y}^t)^2}{(\sigma^t)^2} \tag{11}$$

where $\mathbf{b}_{n,x}^t, \mathbf{b}_{n,y}^t$ stands for the $x, y$ coordinates of the sample point $\mathbf{b}_n^t$, and $\sigma^t$ is the renderer parameter roughly in control of the blur of the rendering output.

As an effect of this rasterizing procedure, the pixel intensity can be arbitrarily large. To normalize it to be always inside $[0, 1]$, we apply a max-normalization followed by a parametrized tanh function to get per-stroke renderings $x_t$ (this corresponds to the $x_t$ introduced in the **Rendering Module** of § 2.1):

$$\bar{\bar{x}}_t = \text{maxnorm}(\tilde{\tilde{x}}_t) = \frac{\tilde{\tilde{x}}_t}{\max(\tilde{\tilde{x}}_t)} \tag{12}$$

$$x_{hw}^t = \text{normalize\_stroke}(\bar{\bar{x}}_{hw}^t; \omega_0^t) = \tanh\left(\frac{\bar{\bar{x}}_{hw}^t}{\omega_0^t}\right) \tag{13}$$

The max-normalization divides each image's pixel values by the highest pixel value of that image, normalizing all pixels to the range $[0, 1]$. Both steps here are important because with just the max-norm, the maximum pixel value of each image is always 1, which is usually not preferred. Conversely, with just the tanh-normalization, $\omega_0^t$ would be required to vary in a much greater range for the output image to look as intended, as $\max(\tilde{x})$ can range from tens to thousands.

## B.2 Compositing Module

With the image pixel value of an individual stroke being in $[0, 1]$, an element-wise sum of all strokes' rendering could still be larger than 1. Hence, another parametrized tanh function is used to get the canvas-so-far at time $t' \in [1, \ldots, T']$ (using $t'$ to index the per-stroke-renderings from Eq. (13) to avoid clashes):

$$x_{\leq t} = \bigotimes_{t'=1}^{t} x_{t'} \tag{14}$$

$$x_{hw}^{\leq t} = \text{normalize\_canvas}([x_{hw}^{t'}]_{t'=1}^{t}; \omega_1) = \tanh\left(\frac{\sum_{t'=1}^{t} x_{hw}^{t'}}{\omega_1}\right) \tag{15}$$

where $x_0$ is the initial blank canvas.

This is an implementation detail underneath the simplified description of the compositing procedure, $x_{\leq t} = x_{<t} \otimes x_t$, appeared in § 2.1; This eschews an accumulative brightening effect that happens when the tanh normalization is applied to the canvas-so-far multiple times (once at every step), as would be required in the procedure in the simplified description.

## B.3 Neural Network Configurations

DooD and AIR in our experiments share the overall neural components.

Convolutional neural nets (CNN) are used as feature extractors for images (one for canvas-so-far, attention window, target, another for residual and its counterparts). Each CNN is equipped 2 Conv2d layers with 3x3 kernel and stride 1 followed by a 1-layer MLP that outputs 256-dim features. The Conv2d layers goes from 1 to 8, then to 16 channels. Notably, we don't use any Max Pooling layer to avoid the spatial-invariant property.

All the prior, posterior distribution parameters are output by their respective MLP (results in 6 separate MLPs). Despite varying input, output dimensions, they share the main architecture: 2 256-dim hidden-layers with tanh non-linearity. The renderer parameters as detailed in Appendix B.1 are predicted by another MLP of the same form, but not modelled as latent variables in our implementation. As a result, DooD employs 7 MLPs. For AIR, An MLP is used as the decoder (i.e., renderer) for AIR, with the same configurations as above.

On top of these, GRUs[5] with 256-dim hidden states are employed for the layout and stroke RNNs.

## B.4 Token Model

To fit our model naturally into the hierarchical Bayesian formulation of the character-conditioned generation and the one-shot classification task, we inserted a plug-and-play **token model** to our generative model. With the learned generative and recognition model, we can regard the learned **prior** $p(o, l, s)$ as a high level **type model** $p(\psi)$ and incorporate a token model $p(z|\psi)$, where $\psi, z$ denote $(o, l, s)$, $(o, l, s')$, respectively (only potentially different in $s$ vs. $s'$). We can then consider the learned variational posterior $q(o, l, s|x)$ to be directly on the type variable $\psi$, i.e. $q(\psi|x)$. The token model $p(z|\psi)$ captures the plausible structural variability of various instances of a character (including affine transformations, motor noise; all embodied in $s'$ given $s$). This can either be learned or set by heuristics.

In our experiment, we simply leverage a uniform distribution over a range of affine transformations and a spherical normal distribution for motor-noise. Note that the flexibility of doing this is thanks to the symbolic latent representation that DooD has, while models with distributed latent representations lack. In detail, the motor noise model has a standard Gaussian distribution with mean centered on the

control points and scale $1e - 3$. The affine model uses uniform distribution and has $x, y$ shift value ranging $[-.2, .2]$, $x, y$ scale $[.8, 1.2]$, rotation $[-.25\pi, .25\pi]$, $x, y$ shear $[-.25\pi, .25\pi]$

## C   Training Details

### C.1   Hyperparameters

The model is trained with the Adam [21] optimizer with a learning rate of 1e-3 for the parameters whose gradients are estimated with NVIL [28] and 1e-4 for the rest, neither with weight decay. The intermediate canvas-so-far $x_{\leq t}$ for $t \neq T$ and residual $\Delta x_t$ produced at each step are detached for both DooD and AIR from the gradient graph for training stability, effectively making them not act as a medium for backpropagation-through-time.

$\beta$ **settings.**   For DooD and it's ablations, $\beta = 4$ is used in the loss function, whereas $\beta = 5$ is used on AIR, both tuned on the MNIST-trained across-dataset generalization task. More specifically, starting from $\beta = 1$, different $\beta$'s with $+1$ increments up to $\beta = 6$ are experimented with models on MNIST, with their behaviors changing from using all steps to using fewer strokes than sufficient to reconstruct the image. The values above are chosen from this range by assessing the marginal likelihood, and qualitatively, whether it's using a sufficient yet parsimonious number of strokes.

**NN Parameter initialization.**   The initial parameters of the last layer of the $l$ MLPs are set to predict identity transformations as per [20]. For the Omniglot dataset only, the initial weights for $o$ MLP's last layer is zeroed and the initial bias is set to a high value (e.g. 8) before passing through a sigmoid function for normalization, because otherwise the model would quickly go to using no steps due to the greater difficulty in joint learning and inference on Omniglot.

**Variable initialization.**   At $t = 0$, variables are initialized to fixed values: $h_0^l, h_0^s, o_0, s_0, l_0, x_0$ are assigned vectors of 0's of different dimensions. In other words, these initial variables are not sampled and not accounted in the joint distribution. This initialization is used in all of the training and evaluation.

### C.2   Stochastic Gradient Estimators

One way of learning the parameters of the generative model $\theta$ and the inference network $\phi$ is by jointly maximizing the lower bound of the marginal likelihood of an image $x$, denoting the joint latent variables by $z$:

$$
\begin{aligned}
\log p_\theta(x) = \log \int dz p_\theta(x, z) &= \log \int dz q_\phi(z|x) \frac{p_\theta(x, z)}{q_\phi(z|x)} \\
&= \log \mathbb{E}_{q_\phi} \left[ \frac{p_\theta(x, z)}{q_\phi(z|x)} \right] \geq \mathbb{E}_{q_\phi} \left[ \log \frac{p_\theta(x, z)}{q_\phi(z|x)} \right] \\
&= \mathbb{E}_{q_\phi}[\log p_\theta(z)] + \mathbb{E}_{q_\phi}[\log p_\theta(x|z)] - \mathbb{E}_{q_\phi}[\log q_\phi(z|x)] =: \mathcal{L}(\theta, \phi) \quad (16)
\end{aligned}
$$

A Monte Carlo gradient estimator for $\frac{\partial}{\partial \theta} \mathcal{L}$ is relatively easy to compute by drawing $z \sim q_\phi(\cdot|x)$ and computing $\frac{\partial}{\partial \theta} \log p_\theta(x, z)$ as the model is differentiable w.r.t. its parameters.

Estimating the gradient for $\frac{\partial}{\partial \phi} \mathcal{L}$ is more involved as the parameters $\phi$ are also used when drawing samples from $q_\phi$. To address this, for each step $t$, denote $\omega^t$ all the parameters of the distribution on variables at $t$, $z^t$. The full gradient can therefore be obtained via chain rule: $\frac{\partial \mathcal{L}}{\partial \phi} = \sum_t \frac{\partial \mathcal{L}}{\partial \omega^t} \frac{\partial \omega^t}{\partial \phi}$. Define $\ell(\phi, \theta, z) := \log \frac{p_\theta(x, z)}{q_\phi(z|x)}$, we can write the loss as $\mathcal{L}(\theta, \phi) = \mathbb{E}_{q_\phi}[\ell(\phi, \theta, z)]$, and let $z^t$ be either the continuous or the discrete subset of latent variables in $(l^t, s^t, o^t)$. How to proceed with computing $\frac{\partial \mathcal{L}}{\partial \omega^t}$ depends on whether $z^t$ is discrete or continuous.

**Continuous.**   For continuous random variable $z^t$, we can use the reparametrization trick to back-propagate through $z^t$ [22, 31]. The trick suggest that for many continuous random variables, drawing a sample $z^t$ from the distribution parametrized by $\omega^t$ yields an equivalent result as taking the output of a deterministic function inputting some random noise variable $\xi$ and parameter $\omega^t$, $z^t = f(\xi, \omega^t)$ where $\xi$ is sampled from some fixed noise distribution $p(\xi)$. This results in the estimator: $\frac{\partial \mathcal{L}}{\partial \omega^t} \approx \frac{\partial \ell(\phi, \theta, z)}{\partial z^t} \frac{\partial f}{\partial \omega^t}$.

**Discrete.** For discrete variables such as $o^t$, the reparametrization trick can't be applied. Instead, we resort to the REINFORCE estimator [28, 31], with a Monte Carlo estimate of the gradient: $\frac{\partial \mathcal{L}}{\partial \omega^t} \approx \frac{\partial \log q_\phi(z|x)}{\partial \omega^t} \ell(\phi, \theta, z)$.

This can be derived as follows (denote $\ell(\phi, \theta, z)$ by $\ell(z)$ and $q_\phi(z|x)$ by $q_\phi(z)$ to simplify notation):

$$
\begin{aligned}
\frac{\partial \mathcal{L}}{\partial \omega^t} &= \frac{\partial}{\partial \omega^t} \int q_\phi(z) \ell(z) dz \\
&= \int \left( \frac{\partial}{\partial \omega^t} \log q_\phi(z) \right) q_\phi(z) \ell(z) dz \\
&= \mathbb{E}_{q_\phi(z)} \left[ \frac{\partial}{\partial \omega^t} \log q_\phi(z) \ell(z) \right] \\
&\approx \frac{\partial \log q_\phi(z)}{\partial \omega^t} \ell(z)
\end{aligned}
\tag{17}
$$

This basic form usually results in a high variance, and we can significantly reduce it by using only local learning signals and a structured neural baseline[28]. The former suggests that we can remove the terms in $\ell(z)$ that don't depend on $\omega^t$ without affecting the result, this allows us to substitute $\ell(z)$ with $\ell^t(z) := \log p_\theta(x|z) p_\theta(z^{t:T}) / q_\phi(z^{t:T})$ such that it only uses learning signals dependent on $\omega^t$. The latter suggests subtracting a control variate $b(z^{<t}, x)$, which takes in $x$ and the previous variables $z^{<t}$ detached from the gradient graph, from $\ell^t(\cdot)$. It is trained by minimizing the mean squared error between $\ell^t(\cdot)$ and $b(z^{<t}, x)$, i.e., $\mathcal{L}_b := \mathbb{E}_{q_\phi}[(\ell^t(z) - b(z^{<t}, x))^2]$. This yields an lower-variance estimator used in learning $\frac{\partial \mathcal{L}}{\partial \omega^t} \approx \frac{\partial q_\phi(z^t)}{\partial \omega^t} (\ell^t(z) - b(z^{<t}, x))$. Finally, the learning signal, $(\ell^t(z) - b(z^{<t}, x))$, is centered and smoothed as in [28]. And the final loss function can be written as $\hat{\mathcal{L}} = \mathcal{L} + \mathcal{L}_b$.

## D   Evaluation Details

### D.1   More across-dataset generalization results

Fig. 3 demonstrates each model's performance when trained on MNIST. Here we show an instance of Fig. 3 with additional results for models trained on each of the 5 datasets (except for the baseline that didn't work on that particular dataset). We further present a Fig. 4-style confusion matrix for DooD-EG.

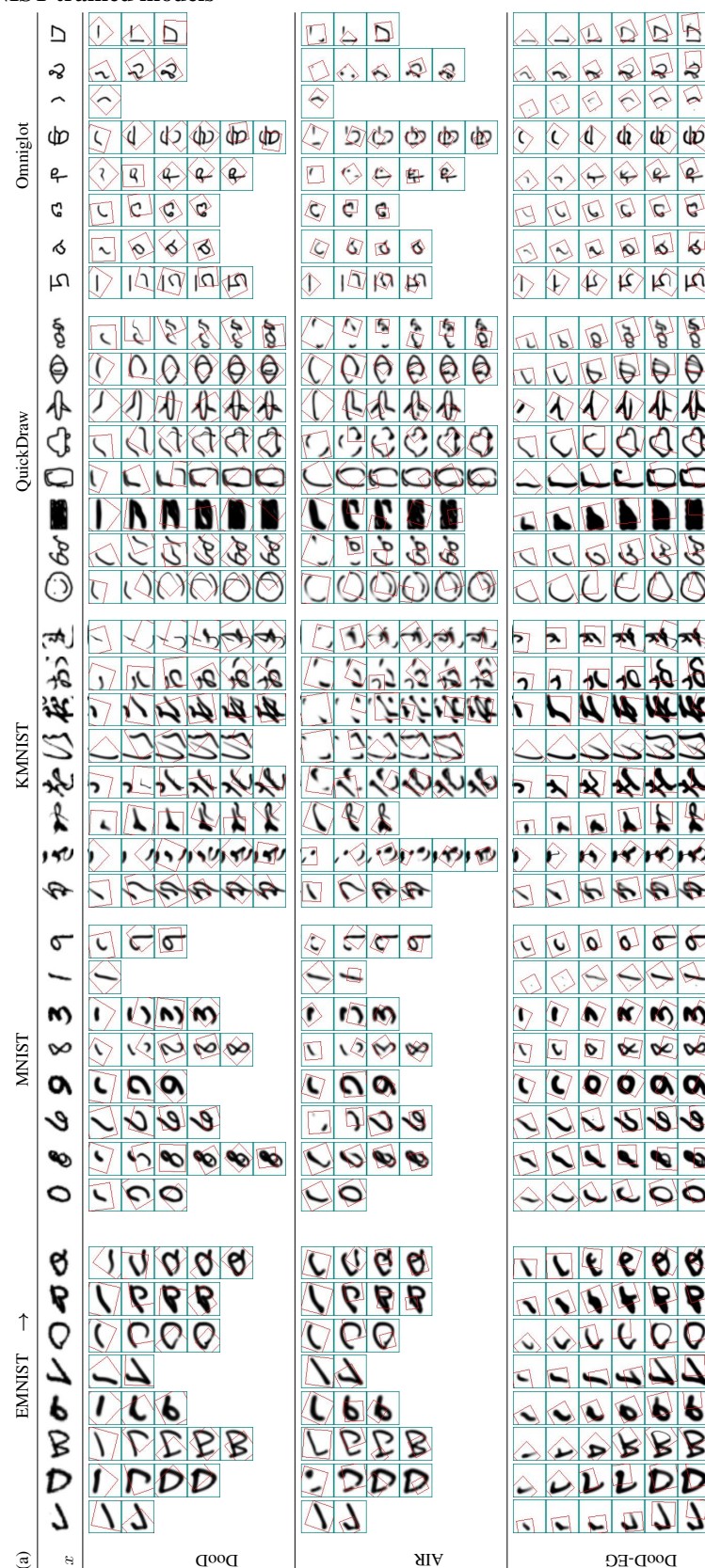

Figure 9: EMNIST-trained model generalize to other datasets.

## D.1.2 KMNIST-trained models

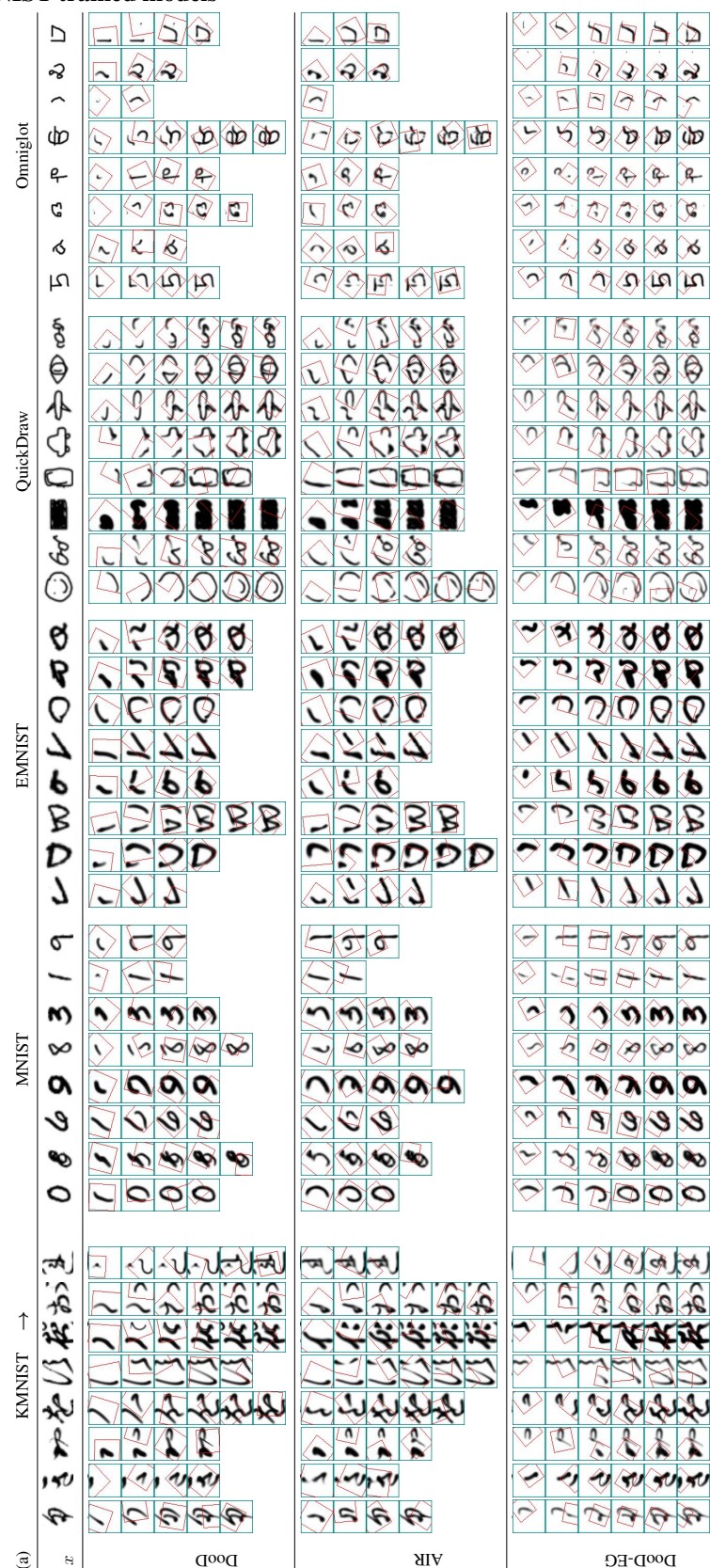

Figure 10: KMNIST-trained model generalize to other datasets.

## D.1.3 MNIST-trained models

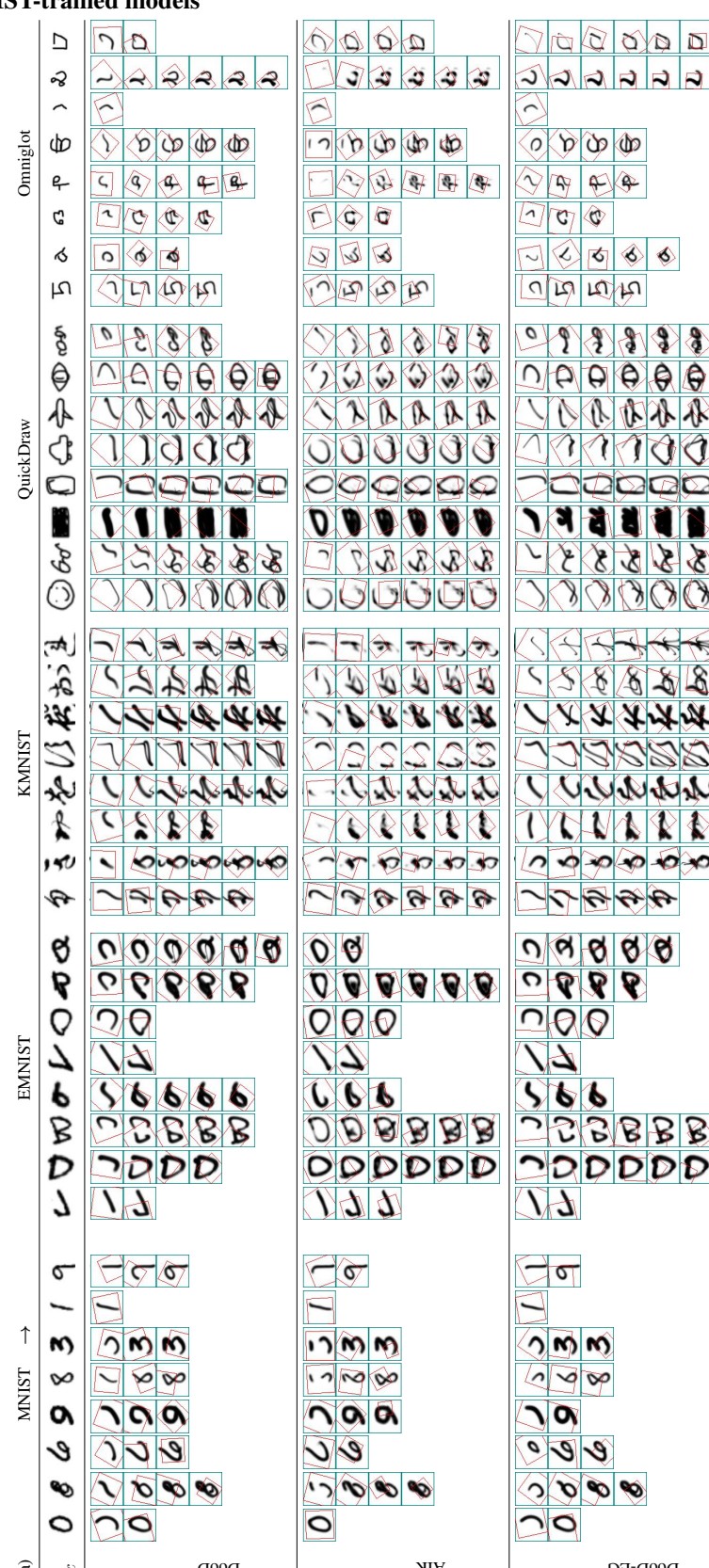

Figure 11: MNIST-trained model generalize to other datasets.

### D.1.4 Omniglot-trained models

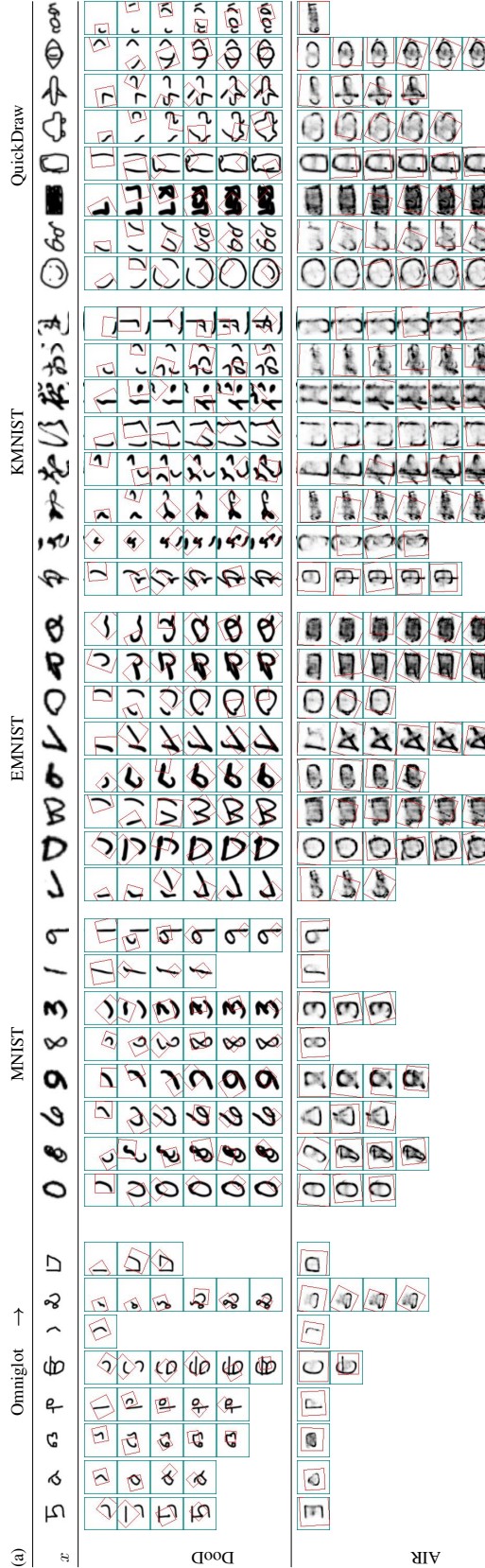

Figure 12: Omniglot-trained model generalize to other datasets.

## D.1.5 QuickDraw-trained models

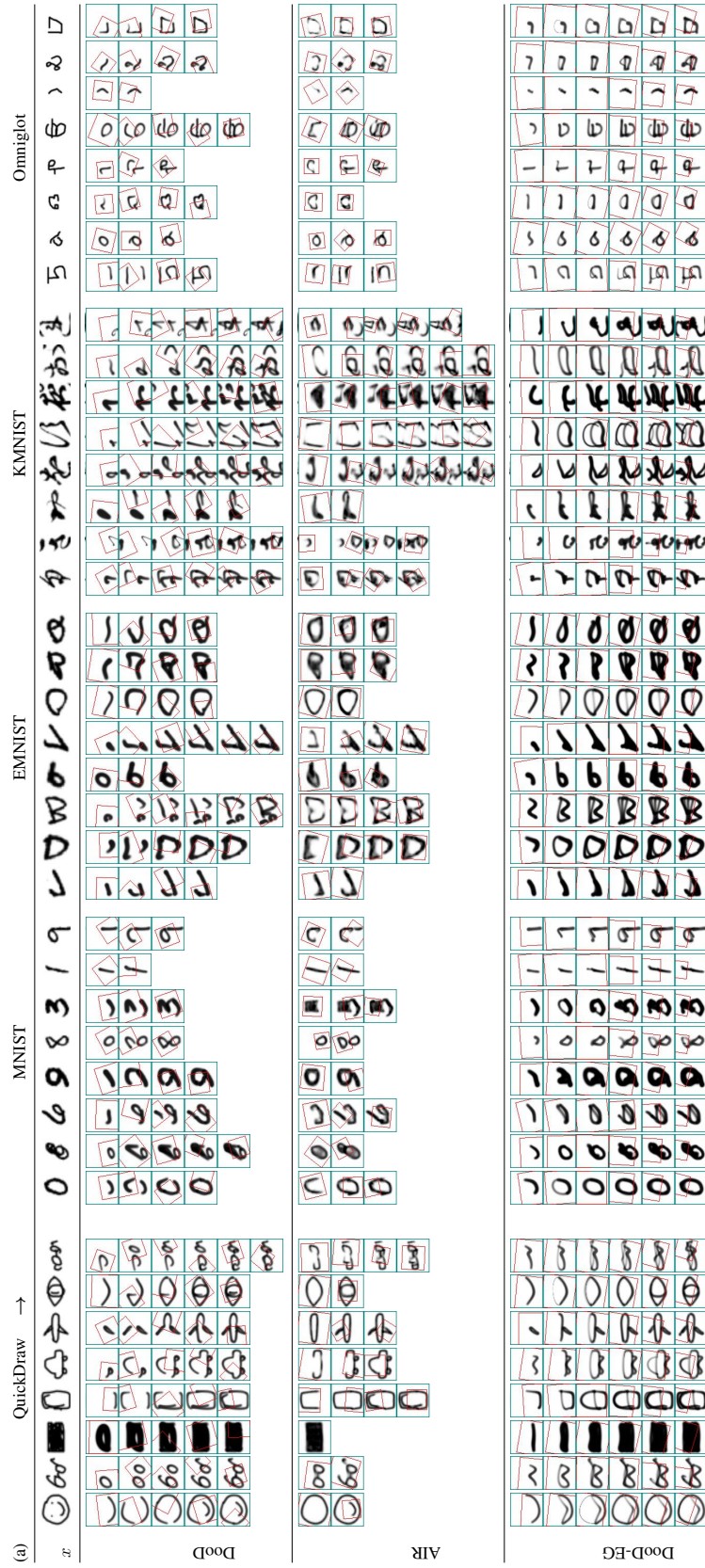

Figure 13: QuickDraw-trained model generalize to other datasets.

### D.1.6  Ablation marginal likelihood evaluation

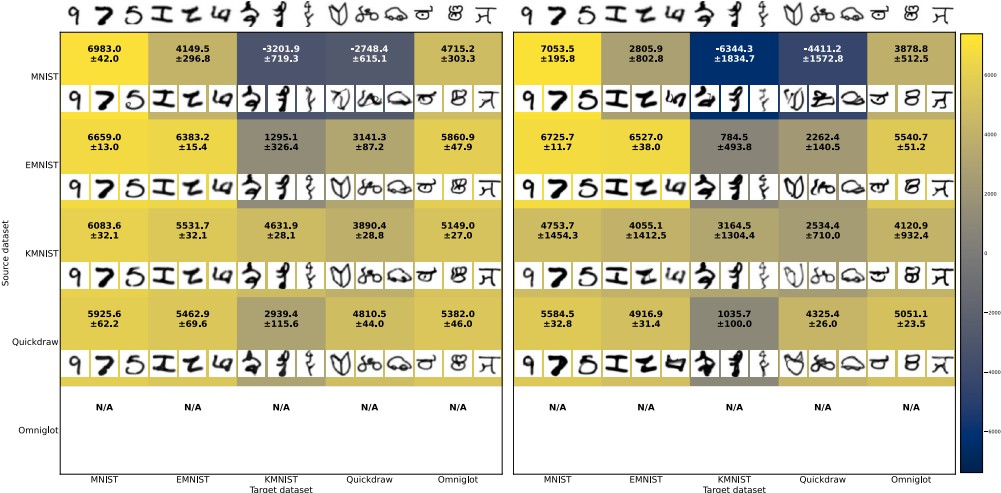

Figure 14: DooD-SP (DooD without sequential prior) and DooD-EG (DooD without execution guidance) cross-dataset log-marginal-likelihood evaluation.

## D.2 More across-task generalization results

### D.2.1 Unconditional generation

Additional unconditional samples from DooD are shown in Fig. 15. In generating these samples, we also make use of the common low-temperature sampling technique [15].

Figure 15: Additional Unconditional generation results from DooD.

### D.2.2 Character-conditioned generation

Additional character-conditioned samples from QuickDraw- and Omniglot-trained DooD are shown in Fig. 16 and Fig. 17.

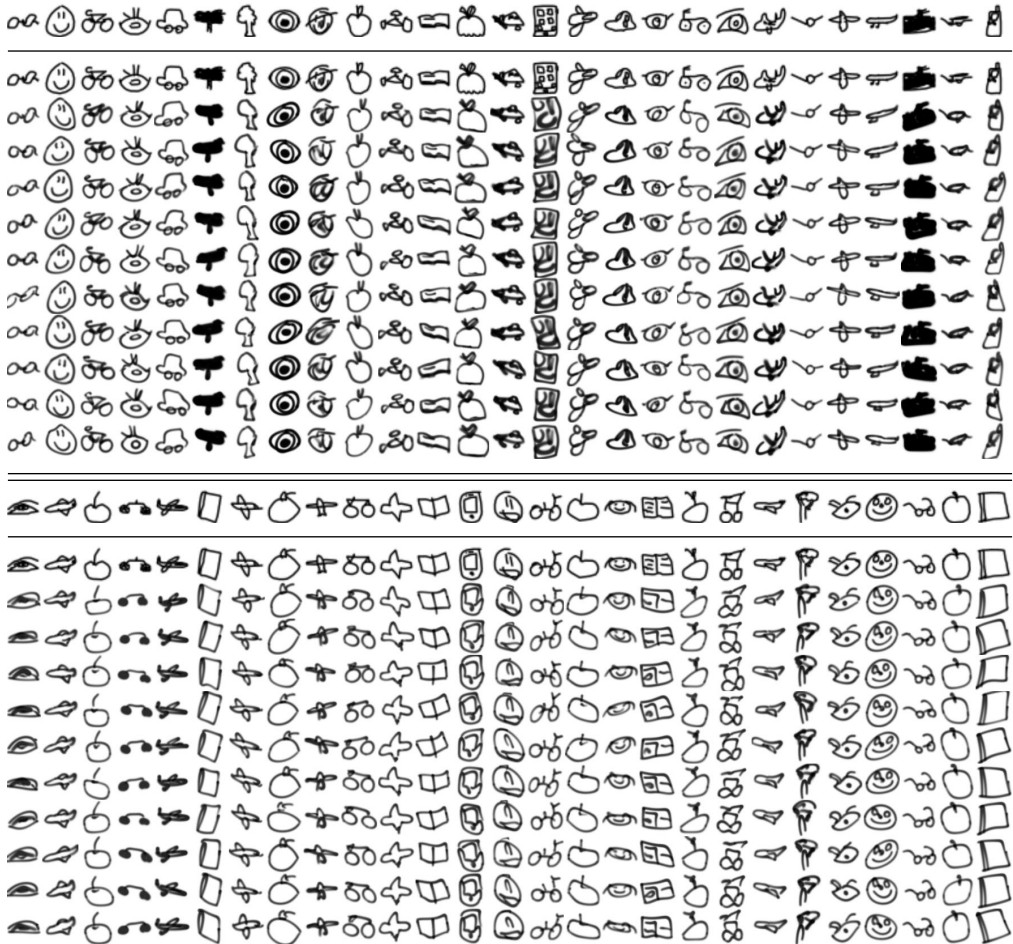

Figure 16: Additional Character-conditioned generation results from QuickDraw-trained DooD.

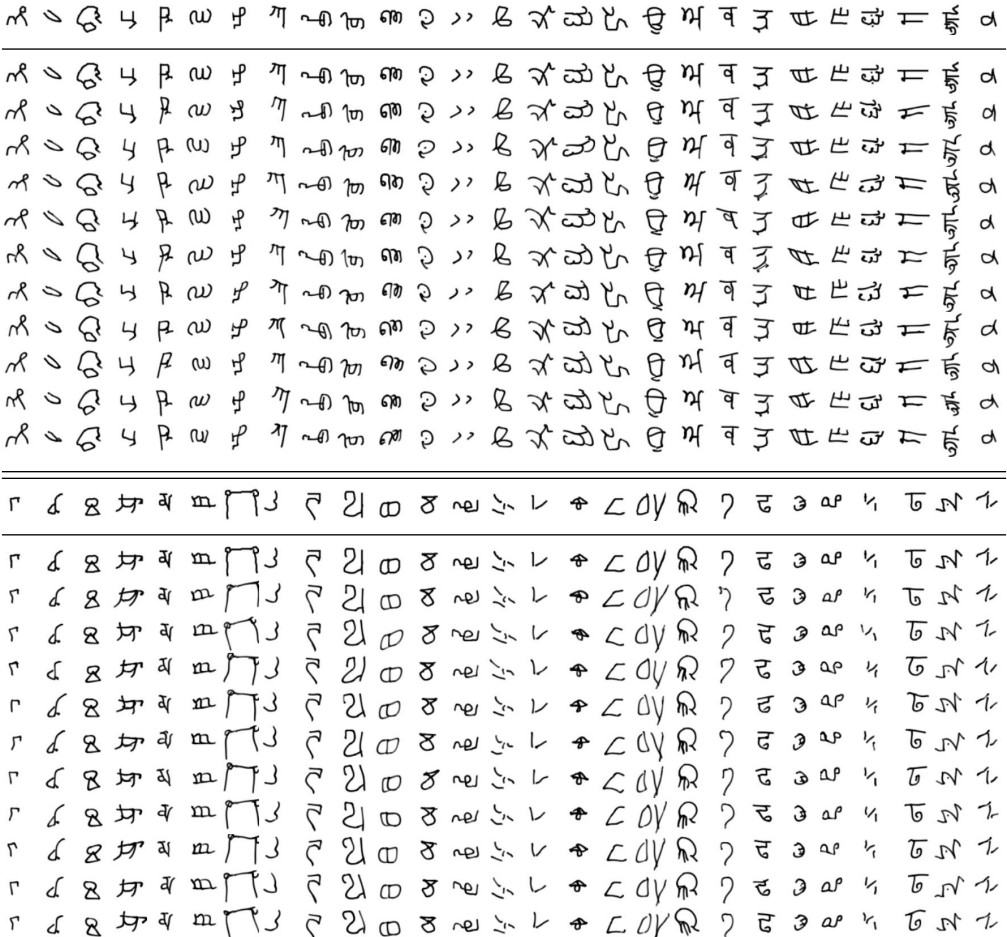

Figure 17: Additional Character-conditioned generation results from Omniglot-trained DooD.

### D.2.3 One-shot classification

DooD performs one-shot classification as follows. When given a support image $x^{(c)}$ from each class $c = [1, C]$, and a query image $x^{(T)}$, it classifies which class $c$ it belongs to by computing the Bayesian score $p(x^{(T)}|x^{(c)})$ for each $c$ and predicting the $c$ with the highest score. The score is computed by:

$$p(x^{(T)}|x^{(c)}) = \int p(x^{(T)}, z^{(T)}, \psi^{(c)}|x^{(c)})d(z^{(T)}, \psi^{(c)}) \tag{18}$$

$$\approx \int p(x^{(T)}|z^{(T)})p(z^{(T)}|\psi^{(c)})p(\psi^{(c)}|x^{(c)})d(z^{(T)}, \psi^{(c)}) \tag{19}$$

$$\approx \sum_{k=1}^{K} \pi_k \max_{z^{(T)}} p(x^{(T)}|z^{(T)})p(z^{(T)}|\psi_k^{(c)}) \tag{20}$$

$$\text{where } \psi_k^{(c)} \sim q(\psi^{(c)}|x^{(c)}), \pi_k \propto \tilde{\pi}_k = p(\psi_k^{(c)}, x^{(c)}) \text{ and } \sum_{k=1}^{K} \pi_k = 1 \tag{21}$$

where $p(z^{(T)}|\psi^{(c)})$ is the plug-and-play token model (as introduced in of Appendix B.4) taking the potential affine transformation, motor noise into consideration. And the $\max_{z^{(T)}}$ is obtained through gradient-based optimization, as in [10, 24].

## E  Limitations

For MNIST-trained DooD in particular, we observe that despite outperforming all baselines in generalization, as evident by the significantly superior mll (Fig. 4), it has a hard time faithfully reconstructing in particular the more complex samples. We attribute this to 2 components of our model that will be investigated in future work, fixing either should significantly improve upon the current generalization performance.

Primarily, we can attribute this to the $s$-component not generalizing strongly. When trained on MNIST, the model rarely sees multiple strokes appearing inside a glimpse. However, this is common in complex dataset such as QuickDraw. This creates a major train/test discrepancy for the $s$-component, causing the model's malfunction—e.g., trying to cover 2 isolated strokes (a "11") with 1 stroke in the middle (a horizontal bar). Despite multiple strokes appearing in glimpses of models trained on other the datasets. This malfunctioning of $s$-component does not happen as much because the model has more robustly learned through its source dataset that it should focus on the center of the given glimpses during inference. A more robust $s$-component design should in principle address this issues.

Fundamentally, however, this can be seen to be caused due to constraints in $l$. By design, $l$ should perfectly segment out each individual stroke and place it into a canonical reference frame, before passing it to the $s$-component—multiple strokes appearing in a single glimpse should not have happened in the first place. Perhaps more flexible STNs[20] could do this (with shear, skew, etc). We expect a combination of the bounding-box approach (as in DooD) and a masking approach (e.g., [4]) might work well, where the masking could help the model ignore irrelevant parts of glimpses before fitting splines to the relevant parts.