# OpenReview forum: "Drawing out of Distribution with Neuro-Symbolic Generative Models"
_NeurIPS.cc/2022/Conference — NeurIPS 2022 Accept_

### Official Review · Reviewer_MGDL · 2022-07-07

**Rating:** 6
**Confidence:** 3
**Soundness:** 3 good
**Presentation:** 3 good
**Contribution:** 3 good

**Summary:**

This paper proposes a generative model for images (drawings or characters) parameterized by sequences of strokes, and a recognition model for inferring the stroke sequence from an image, which are trained jointly by optimizing a variational lower bound.  The generative model makes use of a Bezier-spline stroke model with control points provided by a recurrent convolutional neural network, and a recurrent spatial transformer network used to frame the next stroke.  The method can be used to reconstruct images from character and drawing datasets, generalize across datasets, generate new characters (hence drawing out of distribution), and to perform one-shot character classification.

**Questions:**

* Given the claim that DooD stands out compared to related methods in terms of computational cost, is there any evidence to establish its computational efficiency in practice?
* Does DooD achieve superior one-shot classification compared to other character recognition models based on static images?
* Given that GNS (Feinman and Lake, 2021) can already 'draw out of distribution', what is the motivation for the naming of DooD?
* Line 93: Why is the decision to stop made before composing the current stroke (rather than afterward)?  Doesn't this mean wasting the computational effort to draw the current stroke if it isn't even used?
* Line 96: What is the justification for this Laplace noise model? Why is the scale of tha Laplace fixed? What is the range of the pixel intensities, and is the Laplace distribution appropriate given that range?  Why are the pixel intensities assumed to be conditionally independent given the composited image?  Does the addition of Laplace noise accumulate over time t?  Doesn't this mean that the variance of pixels from earlier strokes is inflated?  Why?
* Equation 6: What is the range for the index t?
* Supplement B.1.  The notation used here (e.g. equations 10-14) differ from the notation used in the paper.  How are the terms in equations 1-6 defined in terms of these quantities?
* Line 119: How is beta tuned?


**Limitations:**

Yes.

**Strengths And Weaknesses:**

## Originality
This method highly resembles other approaches in the literature, e.g. splines used as drawing elements (Mihai and Hare 2013), use of variational objective (Tomczak and Welling 2018), recursive image generation (Feinman and Lake 2021), or use of spatial transformer network within a character recognition model (VHE, Hewitt et al. 2018).  The combination of recursive image generation with a variational objective was previously explored by AIR (Eslami et al. 2016), while the stroke model resembles GNS (Feinman and Lake 2021).  DooD resembles a fusion of AIR and GNS, utilizing a combination of variational objective with a stroke model, and trained on images rather than stroke sequences.  It may be the first recursive image sketching model that uses spatial transformer networks as part of its composition process.
## Quality
Some strengths of the paper are the evaluations on multiple datasets, including cross-dataset transfer, and variety of evaluation tasks (corresponding to the Omniglot challenge).  It is also interesting that the paper investigates the internal representation of the method compared to similar methods.  The contribution of the recurrent structure of the model is demonstrated by an ablation study.  The method qualitatively outperforms baselines in terms of quality of image reconstruction, but the difference in performance is not quantified.

Some weaknesses of the paper are that many of the modeling choices are not well-motivated.  In particular, the choice to use Laplace noise which appears to accumulate over steps (see below for clarity issues) seems unusual and possibly a poor fit to the actual image distribution of human drawings.  The use of only one real baseline (AIR) seems weak, especially since AIR was not developed specifically for line drawings.
## Clarity
The grammar and organization of the paper is good.  However, the details of the method are not described well.  The stroke and rendering components of the model are particularly hard to understand since key details are missing from the main text, and the notations used in the appendix are not defined in terms of the notation in the main text. I am sympathetic to the difficulty of explaining such a complicated method in the page limit provided, so I am personally OK with putting more details in the appendix.  If the clarity of the methods could be improved, I would consider raising my score, which is currently "reject" because of lack of reproducibility.  I am of the opinion that for the paper to be publishable, it needs to contain a complete description of the method details (either in the main text or the supplement), and not just a code submission.  That said, well-documented code would help the reproducibility of the paper even more in conjunction with a more complete description of the method.
## Significance
This paper appears to present a method for modeling human line drawings and handwritten characters from static images which is claimed to have superior cross-domain generalization, computational efficiency and interpretability compared to previous approaches.  It appears that the motivation of the work is to obtain better models of handwritten characters and drawings for computer vision, rather than modeling human drawing.  The interpretability of the method does seem to be superior to AIR, and while methods not based on recursive drawing are not evaluated in terms of interpretability, it is likely that they would suffer from even poorer interpretability.  The method is evaluated against GNS, BPL, and VHE which require stroke-level data.  However, a more relevant comparison would be against one-shot image classification models that are trained on static images.

---

> ### Author Response · Authors · 2022-08-02
> **Comment on the Review by Reviewer MGDL (Part 1)**
>
> Thank you for your review and constructive suggestions.
> We have provided specific responses to your questions and comments below, and have also updated the manuscript with concomitant edits to help clarify the model formulation.
>
> In summary, the update includes:
> 1. Simplifying the model description and formulation;
>     1. clarify purpose of $o_t$ as a ‘continue’ criterion;
>     1. clarify application of the Laplace likelihood at the end of the model unroll;
>     1. explicitly denote the range of indexing variables;
>     1. clarify terminology for execution guidance;
>     1. specify the initial values for different variables involved;
>     1. add a brief description of Bezier curves to Appendix B.1;
> 1. Fix the notation in Appendix B to match the model description in the main text;
> 1. Add more detail on how the hyperparameters are tuned, including for $\beta$, in Appendix C.1;
> 1. Added citations to SCAE and to the introduction to motivate the “what” and “how” factorisation;
> 1. Fix the legend in Figure 3;
> 1. Fix identified typos.
>
> > Originality:
>
> We do agree that DooD can be seen to resemble a fusion of AIR and GNS—in that it brings together the ability of AIR to do amortised variational inference with the ability of GNS to incorporate useful inductive biases into the model. Our aim is to demonstrate that learning a compositional generative model that simultaneously supports efficient (amortized) inference, strong generalization and unsupervised learning is indeed feasible—something that prior work thus far has not demonstrated.
>
> > Quality and Clarity:
>
> We have simplified and clarified the model description to fix things.
> Specifically, the formulation is fixed to show that the Laplace likelihoods do not accumulate over step t, and is only evaluated once at the terminal step determined through latent variable $o_t$.
> At intermediate steps, the composite canvas-so-far is just passed as-is (after detaching from the gradient compute graph) to the next step. This is indeed how our model is also implemented.
>
> > Significance of one-shot classification:
>
> We do indeed select only generative approaches. We do this because our aim is to formulate a single model that can be applied to both across-task and across-data generalisation. Comparing against purely discriminative approaches would not be in the spirit of the fundamental question we are seeking to address. As has been shown previously[1], such methods can indeed outperform generative models, but cannot then be applied in all the settings we wish to evaluate DooD or other generative models in.
>
> [1]: The Omniglot challenge: a 3-year progress report, Lake et al. 2019
>
> > Questions
>
> > Q: "Given the claim that DooD stands out compared to related methods in terms of computational cost, is there any evidence to establish its computational efficiency in practice?"
>
> A: The claim about computational efficiency specifically refers to
> 1. the use of fast amortised inference when compared to the MCMC-based posterior, e.g. as in GNS.
> 2. the ability to generalise to other datasets in a zero-shot manner, i.e. without needing further training.
>
> Specifically for the former, we ran a simple check on the speed of posterior inference and the difference was significant—a forward pass for one image in DooD takes <1s compared to 3-5mins for GNS.
> This was confirmed in correspondence with the GNS authors, and was why we did not run any other formal evaluations.
> Is there something particular you’d like to see us run to further highlight the difference?
>
> –
>
> > Q: "Does DooD achieve superior one-shot classification compared to other character recognition models based on static images?"
>
> A: As summarised by “ The Omniglot challenge: a 3-year progress report” by Lake et. al., some discriminative approaches can achieve high accuracy, but as a consequence of requiring more training data than available, they primarily target between-alphabet classification, as opposed to the originally proposed task of within-alphabet classification, as we study here.
>
> More generally, the aim with DooD is to eschew such task-specific models precisely because they do not apply to other tasks or data as we would like. The same DooD model can perform not just 1-shot classification, but the unconditional/conditional generation tasks, as well as demonstrate the ability to faithfully deconstruct novel observation from entirely different datasets.

---

> > ### Comment · Reviewer_MGDL · 2022-08-02
> > **Thanks for the clarifications and revisions**
> >
> > I will discuss with the other reviewers during the reviewer discussion period, but I'm provisionally raising my score to 6.

---

> ### Author Response · Authors · 2022-08-02
> **Comment on the Review by Reviewer MGDL (Part 2)**
>
> > Q: "Given that GNS (Feinman and Lake, 2021) can already 'draw out of distribution', what is the motivation for the naming of DooD?"
>
> A: GNS demonstrated across-task generalisation (on 4 Omniglot tasks) with the requirement that the generative model is needed to be trained with stroke supervision, and the inference is performed using expensive MCMC-based inference. DooD on the other hand can generalise both across task and across dataset, without needing any supervision.
> We argue that being able to draw the characters and images in a different dataset (say Quickdraw) from what was used for training (say MNIST) is the essence of being able to literally draw out of distribution—hence the naming of the model.
>
> –
>
> > Q: "Line 93: Why is the decision to stop made before composing the current stroke (rather than afterward)? Doesn't this mean wasting the computational effort to draw the current stroke if it isn't even used?"
>
> A: We do this to cover the base case of the model deciding to stop at the very first step. If we allowed the model to compose a stroke in the 1st step before stopping, it wouldn’t be able to generate a blank image.
> Consequently, when deciding to stop at step $t$, the model does not compose a new stroke, and just generates the required parameters for the likelihood (here Laplace).
>
> –
>
> > Q: "Line 96: What is the justification for this Laplace noise model? Why is the scale of the Laplace fixed? What is the range of the pixel intensities, and is the Laplace distribution appropriate given that range? Why are the pixel intensities assumed to be conditionally independent given the composited image? Does the addition of Laplace noise accumulate over time t? Doesn't this mean that the variance of pixels from earlier strokes is inflated? Why?"
>
> A: Our apologies for the confusion.
> We have simplified and fixed the description of the model to address things.
>
> Primarily, the Laplace likelihood is only evaluated at the terminal step, after composing all the intermediate canvases, not at every step as we’d initially formulated it as. The canvas-so-far at each step is simply passed as-is to the next step; there is no additional accumulative noise model.
> We choose Laplace simply because it empirically performs better than the Gaussian, and is indeed a better fit for stroke-style data by virtue of the L1 norm it applies leveraging larger tails, sharpness, and sparsity constraints.
> Its scale is not fixed, but a globally learnt parameter–we have fixed the formulation to show this.
>
> –
>
> > Q: Equation 6: What is the range for the index t?
>
> A: $t$ is not limited in principle. It proceeds until the continuation latent $o_t$ first switches from 1 to 0.
> We have fixed the formulation of the model to reflect this properly.
>
> For the purposes of implementation, we fix the maximum number of steps at train time to $T=6$, and handle any excess steps by appropriate masking.
> The variables with index $t < 1$ are all initialised with a fixed value (0). We did not see any benefit to making them stochastic as they primarily serve to initialise the RNN.
> We have clarified this in the appendix.
>
> –
>
> > Q: Supplement B.1. The notation used here (e.g., equations 10-14) differ from the notation used in the paper. How are the terms in equations 1-6 defined in terms of these quantities?
>
> A: We have updated the appendix to use the same indexing variable as in the main text.
> In the original version, the canvas-so-far in Eq. 14 $(x_{hw}^{T^\prime})$ corresponds to the composition of the rendering of the current stroke and the previous strokes $(x_{<t} \otimes x_t)$.
>
> –
>
> > Q: Line 119: How is beta tuned?
>
> A: This process is explained in line 487 in the original appendix. We have included additional details to Appendix C.1 to help clarify things.
>
> Starting from $\beta=1$, different $\beta$'s with $+1$ increments up to $\beta=6$ are experimented with models on MNIST, with their behaviours changing from using all steps to using fewer strokes than sufficient to reconstruct the image. The values above are chosen from this range by assessing the marginal likelihood, and qualitatively, whether it's using a sufficient yet parsimonious number of strokes.

---

### Official Review · Reviewer_dHqT · 2022-07-09

**Rating:** 6
**Confidence:** 4
**Soundness:** 4 excellent
**Presentation:** 3 good
**Contribution:** 4 excellent

**Summary:**

The paper proposes a neuro-symbolic generative model for stroke based drawing. Unlike prior work in neuro-symbolic generative models, the method doesn’t require explicit supervision. It consists of two parts: 1) A generative model that models the distribution over strokes (sampled from Bezier splines) and their layouts. It is also responsible of rendering the strokes on the canvas in differentiable manner (referred as guided execution). 2) A recognition model that conditions on the target image and infers where to place the strokes. The efficacy of method is demonstrated on two sets of tasks: 1) Generalisation of stroke based reconstruction across datasets. 2) Generalisation across tasks which depict the usefulness of the representation learned.

**Questions:**

- What are the prior probabilities of the initial states of the variables corresponding to layout, stroke, on/off variable and image itself i.e. $p(l_0)$, $p(o_0)$, $p(s_0)$ and $p(x_0)$. Shouldn’t they be a part of Eq. 1?
- (Related to above question) For unconditional generation, how do you sample from the sequential prior? E.g. is the initial state value is sampled from Gaussian distribution?
- What does $\textit{m}$ and $\textit{k}$ correspond to in equation 2 and 3? Are they specific to GMM components weights?
- Could Dth order Bézier splines limit the creation of certain characters/shapes? Or introduce a bias for certain strokes in generation. What is the criteria of selecting a distribution of these splines?
- What does differentiable renderer refer to? Is it just a simple affine transformation or some intricate function of putting control points on the canvas?
- It wasn’t clear what guided execution referred to in the paper. I had to read it again to figure out it’s the differentiable rendering.
- The binary latent variable has some continuous relaxation? The behaviour of Bernoulli distribution (equation 4) can be random in the beginning. Does it affect the training?
- Given the figure 1, shouldn’t the posterior be conditioned on $\Delta x_t$, in L.H.S of equation 6?
- Would a comparison with MWS (https://arxiv.org/pdf/2007.03132.pdf) and another part based generative model (https://arxiv.org/abs/1906.06818) make sense for generalization across datasets experiment? I would recommend to mention the second paper in the related work section.
- Can the transfer of stroke-based reconstructions across datasets be truly called zero-shot as the recognition model have access to the image (x) from OOD dataset? The image from OOD dataset guides the reconstruction (via recognition part of the method) irrespective of whether training was explicitly done on this new dataset or not. This, in my opinion, is one-shot transfer. Curious to know what authors think about this.
- Typo on line 36,  -> model or approaches

**Limitations:**

There doesn't seem to be any obvious negative societal impact of this work.

**Strengths And Weaknesses:**

**Strengths**
- The paper is largely well written and the novelty is clear.
- Shows impressive performance for out-of-distribution characters reconstruction, unconditional generation, conditional generation, and one-shot classification.

**Weaknesses**

Although there are no major concerns, I would appreciate if authors could address the following points.

- A graphical representation of the generative model would make the generative process easier to understand.
- The paper needs to connect certain portions better. E.g how does generative model complements recognition model or vice versa? or how does the shared hidden representations $h_{t}^{l}$ and $h_{t}^{s}$ help recognition and generation?
- The model does not seem to be a fully generative one. I am probably confused here because of equation 1, as characters generation doesn’t seem to be conditioned on the initial states of layout, stroke and on/off variables. See my related question below.
- I liked the transfer of stroke-based reconstructions across datasets experiments. However, I think it’s one-shot transfer instead of zero-shot as the model makes use of the single sample to guide reconstruction. See my related question below.
- Moreover, some terms are introduced without explanations for e.g.  guided execution and bezier splines.

---

> ### Author Response · Authors · 2022-08-02
> **Comment on the Review by Reviewer dHqT (Part 1)**
>
> Thank you for your review and constructive suggestions.
> We have provided specific responses to your questions and comments below, and have also updated the manuscript with concomitant edits to help clarify the model formulation.
>
> In summary, the update includes:
> 1. Simplifying the model description and formulation;
>     1. clarify purpose of $o_t$ as a ‘continue’ criterion;
>     1. clarify application of the Laplace likelihood at the end of the model unroll;
>     1. explicitly denote the range of indexing variables;
>     1. clarify terminology for execution guidance;
>     1. specify the initial values for different variables involved;
>     1. add a brief description of Bezier curves to Appendix B.1;
> 1. Fix the notation in Appendix B to match the model description in the main text;
> 1. Add more detail on how the hyperparameters are tuned, including for $\beta$, in Appendix C.1;
> 1. Added citations to SCAE and to the introduction to motivate the “what” and “how” factorisation;
> 1. Fix the legend in Figure 3;
> 1. Fix identified typos.
>
> ### Commenting on the highlighted weaknesses:
> > "A graphical representation of the generative model would make the generative process easier to understand."
>
> Figure 2 is our effort to depict the generative and recognition processes in finer detail than a standard graphical model typically shows. We intended it in the spirit of explaining how the non-stochastic components (like the STN) also fit in our framework. Is there a particular aspect of the model you feel would be better served by employing a more standard plate formulation? We’d be happy to add it if you think it would be beneficial.
>
> > The paper needs to connect certain portions better. E.g., how does generative model complements recognition model or vice versa? or how does the shared hidden representations $h_t^l$ and $h_t^s$ help recognition and generation?
>
> Beyond the direct relationship between generative and recognition models set up by the ELBO objective, the intermediate renders (i.e. for execution guidance) that are constructed in the generative model are also used when unrolling the recognition model Conversely the recognition model, through the shared hidden states, facilitates the generative model in, e.g., the "partial-completion" task by extracting the hidden states corresponding to the "partial observations" which are otherwise unavailable.
>
> > Moreover, some terms are introduced without explanations for e.g. guided execution and bezier splines.
>
> Thanks for the suggestion. We have added descriptions for “guided execution” in the “rendering module” component of Section 2.1, and for “bezier splines” in Appendix B.1.
>
> ---
>
> > Questions
>
> > Q: "What are the prior probabilities of the initial states of the variables corresponding to layout, stroke, on/off variable and image itself i.e. $p(l_0)$, $p(o_0)$, $p(s_0)$ and $p(x_0)$. Shouldn’t they be a part of Eq. 1?"
>
> A: The variables at $t=0$ are set to fixed initial values, vectors of 0’s for $o_0, s_0, x_0, l_0$, hence are not modelled as random variables or accounted for in computing the joint density. As a result, the $t$ index in Eq. 1 starts at 1.
>
> Additionally, none of the intermediate canvas-so-far values are sampled, only the final canvas-so-far is passed into the image-observation model to parameterize the mean of the Laplace distribution. The initial canvas $x_0$ is initialised to a fixed value of all 0’s to represent a blank canvas.
>
> We have also incorporated the above clarifications in the generative model section of the updated manuscript.
>
> –
>
> > Q: "(Related to above question) For unconditional generation, how do you sample from the sequential prior? E.g. is the initial state value is sampled from Gaussian distribution?"
>
> A: No, the initial states are not treated as random variables themselves, but simply set to a fixed value. We did not see any additional benefit in making them stochastic as these variables are only used to initialise the RNNs, i.e., neither drawn to canvas nor considered in computing the ELBO.
> We’ve added clarification on this initialization in Appendix C.1. of the updated manuscript.
>
> –
>
> > Q: What does $m$ and $k$ correspond to in equation 2 and 3? Are they specific to GMM components weights?
>
> A: $m$ (Eq. 2) is used to index the components over the layout GMM.
>
> $d, k$ (Eq. 3) are used to index spline control points ($d$) and the GMM components ($k$) over each control point respectively.
> A stroke (Dth-order bezier curve) is defined by a list of $D+1$ control points, with a $k$-component GMM corresponding to each control point.
> We have also included clarification of these in Eq. 2, 3 of the updated manuscript.

---

> ### Author Response · Authors · 2022-08-02
> **Comment on the Review by Reviewer dHqT (Part 2)**
>
> > Q: "Could Dth order Bézier splines limit the creation of certain characters/shapes? Or introduce a bias for certain strokes in generation. What is the criteria of selecting a distribution of these splines?"
>
> A: We chose 4th-order splines (using 5 control points) through a simple empirical analysis. Their complexity is sufficient to fit the complexity of strokes typically generated by humans---they can represent complex strokes like loops and whorls quite well.
>
> We do not believe this choice affects the ability to model characters/shapes except in extreme cases (e.g. only 2 control points meaning straight lines).
>
> The complexity of the spline is counterbalanced by the number of steps the model can take. If the model couldn’t draw a particularly complex shape because the strokes were a limiting factor, it is in the model’s interest to figure out how to split this complex stroke into parts that \emph{can} be represented.
> This is directly connected to our characterisation and design of the model in terms of “what” and “how”. It wouldn't limit the ability of the model to reconstruct, just change the complexity of the learned stroke manifold and how that is used.
>
> –
>
> > Q: What does differentiable renderer refer to? Is it just a simple affine transformation or some intricate function of putting control points on the canvas?
>
> A: It is a function that transforms a parametric shape (here splines, through their control points) into a rasterized image, in a differentiable manner. We describe this module in detail in Appendix B.1.
>
> –
>
> > Q: It wasn’t clear what guided execution referred to in the paper. I had to read it again to figure out it’s the differentiable rendering.
>
> A: We have updated the model description to make this clearer. To clarify, it's not the differentiable renderer itself that is referred to as guided execution, but the process of passing the (differentiably) rendered image onto the next step (while also detaching it from the compute graph).
>
> –
>
> > Q: The binary latent variable has some continuous relaxation? The behaviour of Bernoulli distribution (equation 4) can be random in the beginning. Does it affect the training?
>
> A: 1. No, we do not relax this as figuring out how many steps are needed for reconstruction isn't straightforward anymore. We handle the discrete variable using control variates, as is also done in AIR. This was one of the technical challenges when implementing the system—to make proper use of thel NVIL algorithm.
>
> 2. We initialise the parameter of the corresponding neural net's last layer such that it predicts a large value before normalising to become the parameter of the Bernoulli. This helps avoid the model collapsing to not use any steps. We detail this in Appendix B.2.
>
> –
>
> > Q: Given the figure 1, shouldn’t the posterior be conditioned on $\Delta x_t$, in L.H.S of equation 6?
>
> A: $\Delta x_t$ is a deterministically derived variable from the target $x$ and the internally constructed canvas-so-far at step $t$, and not actually observed (in the graphical model sense), which is why it is omitted from the LHS.
>
> –
>
> > Q: Would a comparison with MWS (https://arxiv.org/pdf/2007.03132.pdf) and another part based generative model (https://arxiv.org/abs/1906.06818) make sense for generalization across datasets experiment? I would recommend to mention the second paper in the related work section.
>
> A: MWS is mainly an inference algorithm. We wanted to keep inference the same where possible to investigate the effect of just the modelling changes. MWS/RWS also sets up a tandem objective, which is a pretty big jump from the NVIL that other models use.
>
> For SCAE, we observed that it struggles even on MNIST (as shown in Figure 5(d) of their paper).
> This is why we did not think it would prove an effective comparison or baseline.
> We have also included reference to SCAE in the related work section of the updated manuscript.

---

> ### Author Response · Authors · 2022-08-02
> **Comment on the Review by Reviewer dHqT (Part 3)**
>
> > Q: Can the transfer of stroke-based reconstructions across datasets be truly called zero-shot as the recognition model have access to the image (x) from OOD dataset? The image from OOD dataset guides the reconstruction (via recognition part of the method) irrespective of whether training was explicitly done on this new dataset or not. This, in my opinion, is one-shot transfer. Curious to know what authors think about this.
>
> A: Yes, we believe the task is indeed zero shot as (a) no learning occurs and (b) no additional information is provided beyond the observation.
> Crucially, in comparison to zero-shot classification where it is the label that has never been seen, in this case, the observed data itself is from a completely different domain (MNIST vs Quickdraw for example). Moreover, as with zero-show classification where one relies on the representation similarity of features to classify instances onto new classes, here we rely on the learnt “what” and “how” components to reconstruct instances (from a different domain) onto novel sequences of potentially novel strokes.
>
> –
>
> > Q: Typo on line 36, -> model or approaches
>
> A: Many thanks. We have fixed it in the updated manuscript.

---

### Official Review · Reviewer_xiD8 · 2022-07-11

**Rating:** 2
**Confidence:** 5
**Soundness:** 2 fair
**Presentation:** 4 excellent
**Contribution:** 2 fair

**Summary:**

the work models the drawing, especially, consisting of simple strokes, as a sequential generation problem. The proposed approach leveraged the spatial transformer network to learn and decouple the variance between samples originating from spatial transformations. The generative approach is applied in two task: generating new plots, and one-shot classification.

**Questions:**

How did the authors choose the baseline models to compare? is every model in table 1 generative? Why they are not compared with general classification models?



**Limitations:**

The limitation was discussed outside the main submission. The approach handles a very specific task and the generalization to other tasks are limited.

**Strengths And Weaknesses:**

Strengths
+ plenty of qualitative results and benchmark results are presented. the advantage is highlighted clearly

Weakness:
- AIR is a baseline from 2016, which is pretty outdated. The method proposed seems to be like an STN version of AIR, or in other way, a sequential version of STN.
- The impact of the work seems pretty limited. STN and AIR both explored if they could apply to more realistic dataset like image classification, object detection, etc. while the proposed approach so far proved to only work on stroke-based drawing.

---

> ### Author Response · Authors · 2022-08-02
> **Comment on the Review by Reviewer xiD8**
>
> Thank you for your review.
> We have provided specific responses to your questions and comments below, and have also updated the manuscript with concomitant edits to help clarify the model formulation.
>
> Weakness:
>
> DooD is not “like an STN version of AIR or, in another way, a sequential version of STN”.
> AIR is already that---it fundamentally involves a sequence of steps using STNs.
>
> In so far as DooD bears some similarity to the general framework of AIR, the distinction is non-trivial as enforcing the right inductive biases through the parametric spline latent space, differentiable renderer, and modelling choices in the prior, comes with several methodological and engineering challenges.
> Most crucially, AIR performs poorly on data generalisation, character-conditioned generation, and 1-shot classification, and simply \emph{cannot} be used to perform partially-conditioned generation and unconditional generation, whereas DooD can, and does so very wel.
>
> Secondly, DooD does not claim to be a generic model for all computer vision tasks---in fact its central premise is that judicious modelling of inductive biases for a domain allow it to be more robust and generalisable across that domain than supposedly more general-purpose approaches!
> As we show in comparison to GNS, this also has the additional benefit of being fast and able to learn without the need for additional expensive supervision.
>
>
> > Q: How did the authors choose the baseline models to compare? is every model in table 1 generative? Why they are not compared with general classification models?
>
> A: We chose the baselines based on their relevance to each task we have in our evaluation.
> Regarding table 1 on one-shot classification, yes, every model is generative. As summarised by “ The Omniglot challenge: a 3-year progress report” by Lake et. al., despite some discriminative approaches achieving high accuracy, they usually require much more samples for training and therefore are usually trained and evaluated on between-alphabet classification, as opposed to the originally proposed, more challenging task of within-alphabet classification, as we study here.
>
> More generally, the aim of our model is to demonstrate the strong generalisation ability both across-dataset and across-tasks, which is usually not attainable with general classification models (discriminative models).

---

> > ### Author Response · Authors · 2022-08-07
> > **Followup comment**
> >
> > To further elaborate on our previous comments, our baselines are motivated from the perspective of solving the full Omniglot challenge which is to learn everything about the domain that a human can. This *requires* a model to generate as this is a key characteristic of what humans do with writing/drawing. Given an exemplar of a character or shape, they can draw other instances of it. They can also infer the process (“how”---strokes and their ordering, albeit not perfectly) by which it was drawn, and be able to do all this with less data and minimal, if any, supervision. This is our rationale for focussing on generative models.
> >
> > Having said that, we are happy to (re-)run some discriminative models that are applicable to the within-alphabet one-shot classification task here for the final version of the paper if it is accepted, and if the reviewer and AC think that that would improve the paper.

---

### Official Review · Reviewer_uLsf · 2022-07-15

**Rating:** 7
**Confidence:** 3
**Soundness:** 4 excellent
**Presentation:** 3 good
**Contribution:** 3 good

**Summary:**

- The paper proposes an unsupervised neuro-symbolic generative model for general-purpose pen-drawing/doodling frames as a sequential composition of strokes.
- Analyze the model and explore its benefits in terms of interpretability and out-of-distribution generalization.


**Questions:**

- **AIR**: The first paragraph talks about decomposing to “what” and “how” etc. I think the AIR paper should be cited at this point to give it the credit of originating these ideas.
- **Affine transformations**: Is it possible to decompose the doodles to general sub-curves that will be directly sampled relatively to the positions of the previous strokes, such that there won’t be a need to apply transformations on each of them? If I understand correctly, it may help simplify the model.
- **Related work**: I recommend moving the related work section to the beginning of the paper after the intro rather than before the end.
- **Figure**: In figure 3 right it’s unclear to me why it says MNIST often has 10 strokes while quickdraw has 2? Are these labels swapped or I misunderstood something?


**Limitations:**

Yea but only in the supplementary. I recommend making at least a brief version in the main paper.

**Strengths And Weaknesses:**

**Strengths**
- **Technical details**: The paper does a great job in describing the model in a formal, precise and detailed manner.
- **Writing quality**: The writing quality is good and the model figure is clear and helpful.
- **Experiments**: The paper extensively covers a variety of experiments measuring generalization across both different tasks and datasets, provides a lot of helpful visualizations and also performs ablations.
- **Unsupervised**: The model achieves SOTA performance compared to other neuro-symbolic approaches even though it doesn’t rely on strong supervision or augmentations of the data.
- **Out of distribution generalization**: The general-purpose framing of the approach allows it to generalize out of the training distribution (MNIST) to new characters (Japanese) and even doodles (car, bike) and to a greater number of strokes. It does better than AIR.
- **Interpretability**: The model generates doodles in a sequential easy-to-visualize manner, decomposing them in mostly a natural manner – based on looking at the few shown examples, similar in some cases to how a person might approach the drawing task (although in other cases the decompositions are less likely).
- **Generative capabilities**: The model can create new examples given a new doodle, and can complete a partial doodle into diverse results.

**Weaknesses**
- **Task**: The model explored by the paper is very specific to the particular task (decomposing doodles to strokes, formulating a stroke as a controlled curves that could be adjusted through spatial transformations), which is potentially a bit narrow in itself. It’s unclear to me what is the potential of the approach to scale to broader domains. It could be interesting to explore visual compositionality in a more general manner, but that’s not studied in the paper.
- **Model complexity**: the model is quite complex, with 5 different models and subparts:  having generation and recognition models, with the former consisting of sampling the layout, then the strokes, then performing an affine transformation on them (rendering), then compositing them. I think it would be good to both add more intuitive explanation about this process, its structure, and why it is a good way to approach the task. And if there could be simplifications to the approach that would also be great.
- **Idea presentation**: I think the presentation could be improved to highlight more the big picture, the motivation and the takeaways. A good amount of background context, related work and overall motivation are being discussed, as well as the limitations and ethical considerations.

Considering both strengths and weaknesses, I originally was in terms of scoring between 6 and 7, leaning to 7 due to technical contribution and formal presentation and to 6 due to the potentially bit narrow scope of the task, but then due to the great evaluation section I converged at 7.

---

> ### Author Response · Authors · 2022-08-02
> **Comment on the Review by Reviewer uLsf**
>
> Thank you for your review and constructive suggestions.
> We have provided specific responses to your questions and comments below, and have also updated the manuscript with concomitant edits to help clarify the model formulation.
>
> > Q: AIR: The first paragraph talks about decomposing to “what” and “how” etc. I think the AIR paper should be cited at this point to give it the credit of originating these ideas.
>
> We have added references to the relevant work, including AIR, that motivates our perspective on the problem.
>
> Note however, that AIR describes things in terms of “what” and “where”, not specifically “how”.
> The difference is subtle, but we believe, pretty important. The question of “how” is fundamentally one of global \emph{process}---in this case, a specific sequence of “where” choices. Looking at a set of independent “where” choices, as AIR does (see Fig2 [left] in AIR) does not help capture correlations in the sequences of strokes one might observe. This is further reflected in our model by the separate mechanisms (RNNs) for the stroke and layout.
>
> –
>
> > Q: Affine transformations: Is it possible to decompose the doodles to general sub-curves that will be directly sampled relatively to the positions of the previous strokes, such that there won’t be a need to apply transformations on each of them? If I understand correctly, it may help simplify the model.
>
> If we have understood the question correctly, are you commenting on each stroke depending on the composite canvas rather than a relative transformation from where the previous stroke was drawn?
>
> If so, it is conceptually true that one may model strokes in this manner, but doing so would complicate matters elsewhere in the model. The issue is primarily one of being able to render onto a finite canvas.
> Choosing to model strokes in relative manner would still necessitate keeping track of its actual global layout because we would need to constrain things to stay within the image boundaries. If this did not happen, there would not be any learning signal through the likelihood as strokes outside the image would not get rendered.
>
> Modelling strokes as depending on separate transforms over the canvas-so-far helps avoid this issue, while also allowing for simpler global constraints on the transforms themselves, by not needing to be adapted relative to other strokes.
>
> –
>
> > Q: Related work: I recommend moving the related work section to the beginning of the paper after the intro rather than before the end. In figure 3 right it’s unclear to me why it says MNIST often has 10 strokes while quickdraw has 2? Are these labels swapped or I misunderstood something?
>
> Thank you for the suggestion; we will consider structuring the sections once we’ve analysed how it affects the narrative flow of the manuscript. We currently have it at the end to better focus on the primary problem statement and our proposed approach.
>
> Thank you also for spotting the label error. This has now been fixed.

---

### Author Response · Authors · 2022-08-02
**General comment on the Reviews**

We thank all the reviewers for their comments and valuable suggestions.

We’d like to emphasise that DooD demonstrates a generative approach to visual concept learning that has efficient (amortised) inference, strong generalisation and is learnt unsupervised.
As discussed in Lake et. at.’s Omniglot challenge 3-year progress report, the evaluations on the “seemingly simple” across-task generalisation has been extremely challenging for neural approaches.
DooD generalises well not just across task, but across dataset as well, making it an important step forward towards more human-like visual concept learning.

We have replied to each review in detail and updated the manuscript to clarify the questions raised in the reviews.

In summary, the update includes:
1. Simplifying the model description and formulation;
    1. clarify purpose of $o_t$ as a ‘continue’ criterion;
    1. clarify application of the Laplace likelihood at the end of the model unroll;
    1. explicitly denote the range of indexing variables;
    1. clarify terminology for execution guidance;
    1. specify the initial values for different variables involved;
    1. add a brief description of Bezier curves to Appendix B.1;
1. Fix the notation in Appendix B to match the model description in the main text;
1. Add more detail on how the hyperparameters are tuned, including for $\beta$, in Appendix C.1;
1. Added citations to SCAE and to the introduction to motivate the “what” and “how” factorisation;
1. Fix the legend in Figure 3;
1. Fix identified typos.

---

### Meta-Review · Area_Chair_FiAA · 2022-08-27

**Recommendation:** Accept
**Confidence:** Certain

**Metareview:**

The paper presents DooD, an unsupervised approach for stroke-based generation of line drawings and handwritten characters, which the authors show outperforms other methods in generalization and interpretability. The reviewers agree that this is an innovative and useful contribution, and is well-expressed in the paper. Some concerns are raised related to the insights that one can draw from this domain to other forms of image generation; however, these objections are not fatal, in my opinion. Note that in a private communication with me, the only reviewer not recommending acceptance wrote: "I would keep my score, though I won't be disappointed if the paper will be accepted."

I recommend acceptance of the paper.

**Award:**

No

---

### Decision · Program_Chairs · 2022-09-14

Accept